# Mapping the conformational energy landscape of Abl kinase using ClyA nanopore tweezers

Fanjun Li[1], Monifa A. Fahie[2], Kaitlyn M. Gilliam[2], Ryan Pham[1] & Min Chen [1,2 ✉]

Protein kinases play central roles in cellular regulation by catalyzing the phosphorylation of target proteins. Kinases have inherent structural flexibility allowing them to switch between active and inactive states. Quantitative characterization of kinase conformational dynamics is challenging. Here, we use nanopore tweezers to assess the conformational dynamics of Abl kinase domain, which is shown to interconvert between two major conformational states where one conformation comprises three sub-states. Analysis of kinase-substrate and kinase-inhibitor interactions uncovers the functional roles of relevant states and enables the elucidation of the mechanism underlying the catalytic deficiency of an inactive Abl mutant G321V. Furthermore, we obtain the energy landscape of Abl kinase by quantifying the population and transition rates of the conformational states. These results extend the view on the dynamic nature of Abl kinase and suggest nanopore tweezers can be used as an efficient tool for other members of the human kinome.

[1] Department of Chemistry, University of Massachusetts Amherst, Amherst, MA 01003, USA. [2] Molecular and Cellular Biology Program, University of Massachusetts Amherst, Amherst, MA 01003, USA. ✉email: mchen1@chem.umass.edu

Protein kinases are attractive drug targets because they play a central role in regulating the majority of cellular pathways by catalyzing the phosphorylation of target proteins involved in complex physiological processes[1]. Abelson (Abl) kinase is a cytoplasmic tyrosine kinase. Dysregulated Abl kinase activity has been linked as a contributing factor in chronic myeloid leukemia (CML). Imatinib is an ATP-site inhibitor approved by FDA to treat CML[2,3]. However, the presence of mutations in advanced-stage patients render the drug ineffective[4–6], which has motivated the development of new therapeutics by modifying imatinib structure or exploring of alternative druggable states[7,8].

Abl kinase has a conserved catalytic kinase domain, consisting of a N-terminal lobe (N-lobe) and a C-terminal lobe (C-lobe) that are linked by a flexible hinge. ATP binds to the cleft between two lobes while the substrate protein/peptide binding site is located mainly at the C-lobe (Fig. 1a)[9,10]. Crystal structures of Abl kinase domain at different ligand-bound states mainly differed in the arrangement of four key elements including the activation loop (A-loop), the Asp-Phe-Gly (DFG) motif, the glycine-rich loop (G-loop), and the αC-helix (Fig. 1a)[11,12]. Biological functions of kinases are dictated by their structural flexibility switching between one active and multiple inactive states[13]. Intermediate conformations emerging during state transitions could be potential druggable states[14]. Therefore, a comprehensive understanding of Abl conformational energy landscape, including structures, the relative propensity of these states, and state transition kinetics, are desired.

Despite a wealth of ligand-bound crystal structures of Abl are available, no crystal structure of apo Abl has been solved to date, which limits our understanding of possible conformations adopted by apo Abl. Hydrogen exchange mass spectrometry (HX MS)[15] and nuclear magnetic resonance (NMR) studies[11,16,17] suggested that apo Abl kinase is intrinsically flexible. Computational simulations have successfully identified some metastable states of apo Abl kinase[12,18], but it remains challenging to characterize the large conformation changes occurring over long timescales (μs ~ ms) owing to the large protein size and the small calculation timesteps (~2 fs)[19–21]. A recent NMR study performed at 10 °C, has identified three structural conformers of apo Abl kinase domain and the conformational landscape[22]. Notably, the structures determined at the relatively low temperature may not reflect all physiologically relevant conformations.

Although NMR is a powerful technique for analysis of structural dynamics at the various temporal resolutions, it usually requires a large volume of isotope-labeled samples in high concentrations[23]. Also, signals of sparsely or transiently populated intermediates may be obscured in ensemble measurements. These limitations motivated us to explore a single molecule nanopore approach, as a complementary technique to NMR for studying the conformational dynamics of biological molecules. Nanopore technology has shown great success in DNA sequencing[24,25], small molecules, and protein identification[26–30], is emerging as an attractive single-molecule tool for probing protein conformational dynamics[31,32]. Apart from the general strengths of single-molecule techniques such as the ability to access conformational heterogeneity, transient or sparsely populated intermediates, and sequential steps of enzyme's catalytic cycle[33–37], nanopore tweezers own some unique advantages including requirements of small amounts of protein samples (~picomoles and nanomolar concentration in flow cells), label-free and capable of detecting structural dynamics in a wide range of timescales (μs ~ mins)[38]. Previous works showed that natively folded proteins[31,39–42] can be reversibly trapped with Cytolysin A (ClyA) nanopore variants to reveal their ligand bound and catalytic states. Notably, four different ionic states of dihydrofolate

reductase (DHFR) were detected[39] and three anomeric maltose-bound states of maltose-binding protein (MBP) were resolved by ClyA nanopores, the conformational difference among which was indicated however was not explicitly revealed by any structural studies[31]. Together, these works demonstrate the high spatial sensitivity of ClyA nanopore tweezers for resolving conformational states with subtle differences within flexible enzymes.

In this work, we apply ClyA nanopore tweezers to reveal a comprehensive conformational energy landscape of the Abl kinase domain and its interactions with ligands. Our work also paves the way for the further development of nanopore tweezers as an emerging transformative technology in fundamental research and medical applications.

## Results and discussion

**Nanopore tweezers reveal multiple states of apo Abl kinase.** Current recording experiments with ClyA-AS nanopore tweezers were performed on the catalytic domain of Abl kinase to access the kinase conformational ensembles (Fig. 1b). Wildtype Abl kinase domain (hereafter Abl) was driven to enter the nanopore by electroosmotic flow from *cis* to *trans*, resulting in a reduction in the ionic current (Fig. 1c). Residual current ($I_{res}$) was calculated from blocked pore current ($I_B$) and open pore current ($I_O$) with $I_{res}\% = 100\% \times I_B/I_O$ (Supplementary Fig. 1). Abl can be reversibly trapped under the applied potential of −80 mV with a mean trapping time ($\tau_{trapping}$) of $0.06 \pm 0.03$ s ($N = 5$, $n = 1068$; $N$ indicates pore number, n indicates total event number), which is too short for monitoring the enzyme's conformational dynamics over multiple turnovers as the $k_{cat}$ is $7.1 \pm 0.7$ s$^{-1}$ (Supplementary Fig. 2, Supplementary Table 1), requiring minimal 0.14 s to observe one catalytic cycle.

We then tested the trapping time of Abl under different applied potentials (−80 ~ −120 mV). A voltage-dependent trapping was observed with the longest trapping time shown at −100 mV of $2.33 \pm 0.66$ s ($N = 6$, $n = 661$) (Supplementary Fig. 3, Supplementary Table 2). Besides, two alternating current states (S1 and S2) were observed at all tested potentials (Fig. 1c, Supplementary Fig. 4). However, the trapping time was too short at all tested voltages that not many complete S1 and S2 events were observed (Supplementary Fig. 5), the S1/S2 transition kinetics cannot be determined with the wildtype Abl construct.

To extend the $\tau_{trapping}$, we introduced positively charged tags to Abl to enhance the electrophoretic force[43]. We prepared two Abl constructs with a positively charged peptide tail at its N-terminus, termed $_{N4pos}$Abl, or to its C-terminus, named Abl$_{C4pos}$ (peptide sequence for N4pos tag: KRKKSGG; C4pos tag: GGSKKRK). Under the applied potential of −80 mV, both Abl$_{C4pos}$ and $_{N4pos}$Abl showed their longest $\tau_{trapping}$ of $1.38 \pm 0.14$ s ($N = 3$, $n = 657$) and $21.48 \pm 3.60$ s ($N = 5$, $n = 753$), respectively (Fig. 1d, Supplementary FigS. 3, 6, Supplementary Table 2). We surmised that the two kinase proteins may enter ClyA in different orientations with $_{N4pos}$Abl binding to the ClyA lumen stronger than Abl$_{C4pos}$. Since $_{N4pos}$Abl exhibited the longest trapping time, we decided to study this construct in detail. In current recording experiments, $_{N4pos}$Abl induced similar current signals as Abl (Fig. 1c–e, Supplementary Fig. 7). The current states are characterized by their relative residual current ($I_{res}\%$), state dwell time ($\tau$), and state population (P) (Supplementary Table 3). Apo $_{N4pos}$Abl generated two alternating current states (S1 and S2) with distinct signatures (Fig. 1d–e). The S2 state showed ~95% occupancy and a relative residue current $I_{res}\%$ ranging from 38 to 58%. The less populated S1 state (~5% occupancy) contained three distinct sub-states defined by the $I_{res}\%$: S1a, S1b, and S1c. Notably, the three sub-states did not occur in a random manner. Instead, Abl alternately visited S1b and an S1a/c joint state.

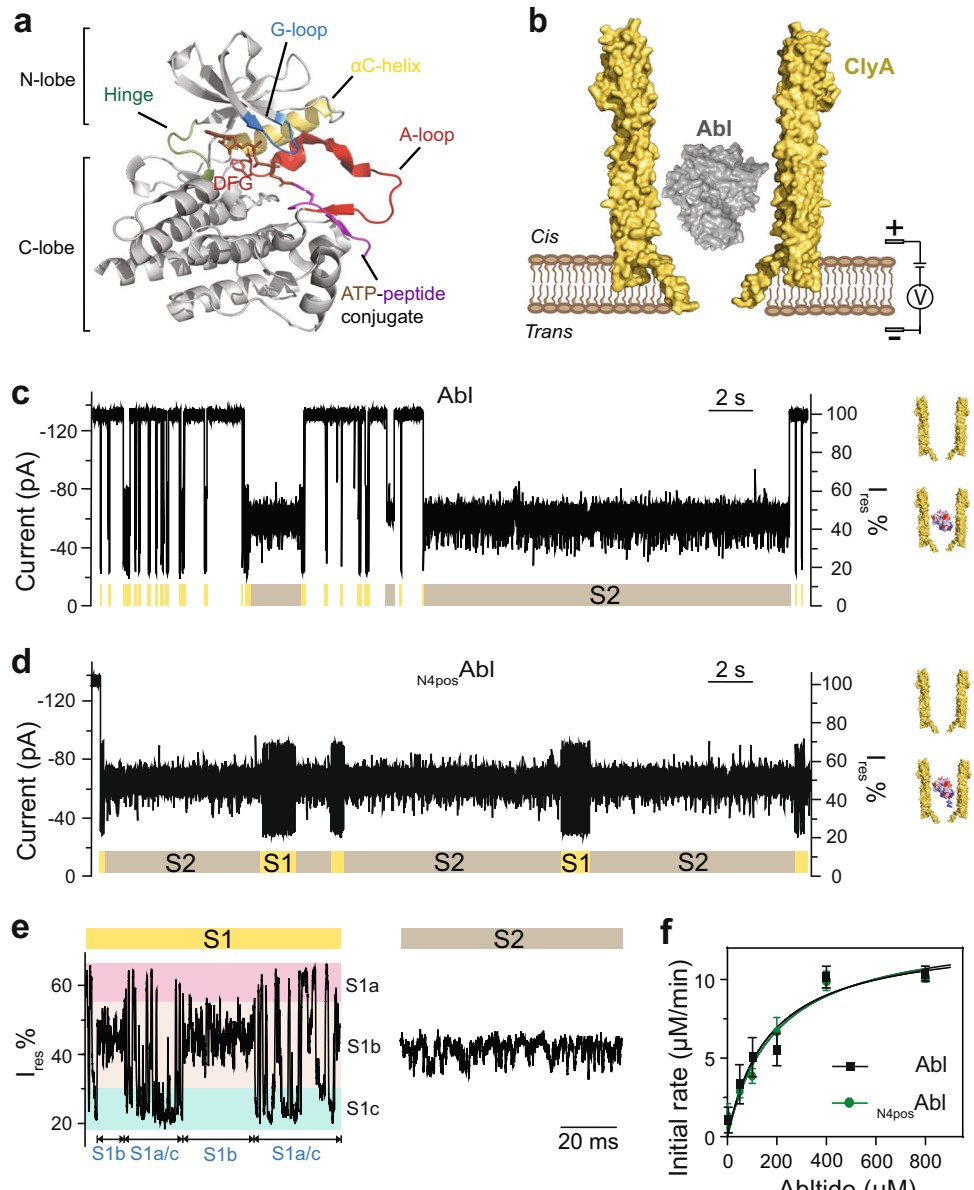

**Fig. 1 Trapping of apo Abl kinases with ClyA nanopore tweezers. a** Crystal structure of the catalytic domain of human Abl kinase (Abl, gray) bound with ATP-peptide conjugate (PDB: 2G1T). ATP in brown and peptide in purple. Structural elements are indicated with different colors, hinge in green, G-loop in blue, αC-helix in yellow, and A-loop in red. Side chain of DFG-motif is shown as stick. **b** Schematic representation of a single Abl kinase (gray) confined in a ClyA nanopore (yellow) embedded in a planar lipid bilayer (brown) with negative applied potential at *trans*. **c**, **d** Representative current traces and schematics of Abl (**c**) and $_{N4pos}$Abl (**d**) trapped in a ClyA nanopore. $_{N4pos}$Abl indicates Abl attached with a positively charged peptide tail at its N-terminus. Peptide sequence for N4pos tag: KRKKSGG. Signal pattern S1 (yellow), S2 (brown) are observed when Abl is trapped within nanopore. **e** Zoomed in trace of S1 and S2 states from $_{N4pos}$Abl trapping signal. Three sub-states were observed within S1, namely, S1a (highlighted with a pink band), S1b (highlighted with a beige band), and S1c (highlighted with a cyan band). The state transition pathways were: S1 ↔ S2, S1a/c ↔ S1b, and S1a ↔ S1c. S1a/c indicates an S1a/c joint state. **f** Michaelis-Menten kinetics analysis of Abl (black) and $_{N4pos}$Abl (green). Data were represented as mean ± SD, $n = 3$ independent replicates. Source data are provided as a Source Data file. The current traces were collected at −80 mV in 150 mM NaCl, 100 mM Tris-HCl, pH 7.5, with ~100 nM Abl or $_{N4pos}$Abl added to *cis* chamber, at 22 °C.

During the S1a/c joint state, Abl was found to oscillate rapidly between S1a and S1c. As a control, the Michaelis-Menten kinetic analysis demonstrated that the N-terminal positive tail had no influence on the catalytic activity (Fig. 1f, Supplementary Table 1).

In short, we obtained an active construct $_{N4pos}$Abl with increased dwell time to monitor the conformational dynamics of a single Abl molecule over extended time. More importantly, we observed that the apo Abl kinase induced multiple current states in ClyA-AS nanopore tweezers.

**S1 sub-states resemble substrate-binding conformations**. We next monitored the current signal of $_{N4pos}$Abl in the presence of various ligands[44]. Firstly, the addition of only MgCl$_2$ did not trigger any noticeable change (Fig. 2a–b, see Supplementary Fig. 8a–b for longer traces), suggesting that the divalent cation did not perturb the ClyA/Abl system. In contrast, re-distributions of S1 sub-states were observed when different substrate/ligands were presented (Fig. 2c–f; Supplementary Table 4; see Supplementary Fig. 8c–f for longer traces): A substrate peptide Abltide binding shifted the current state equilibrium to a pre-existing state S1b by

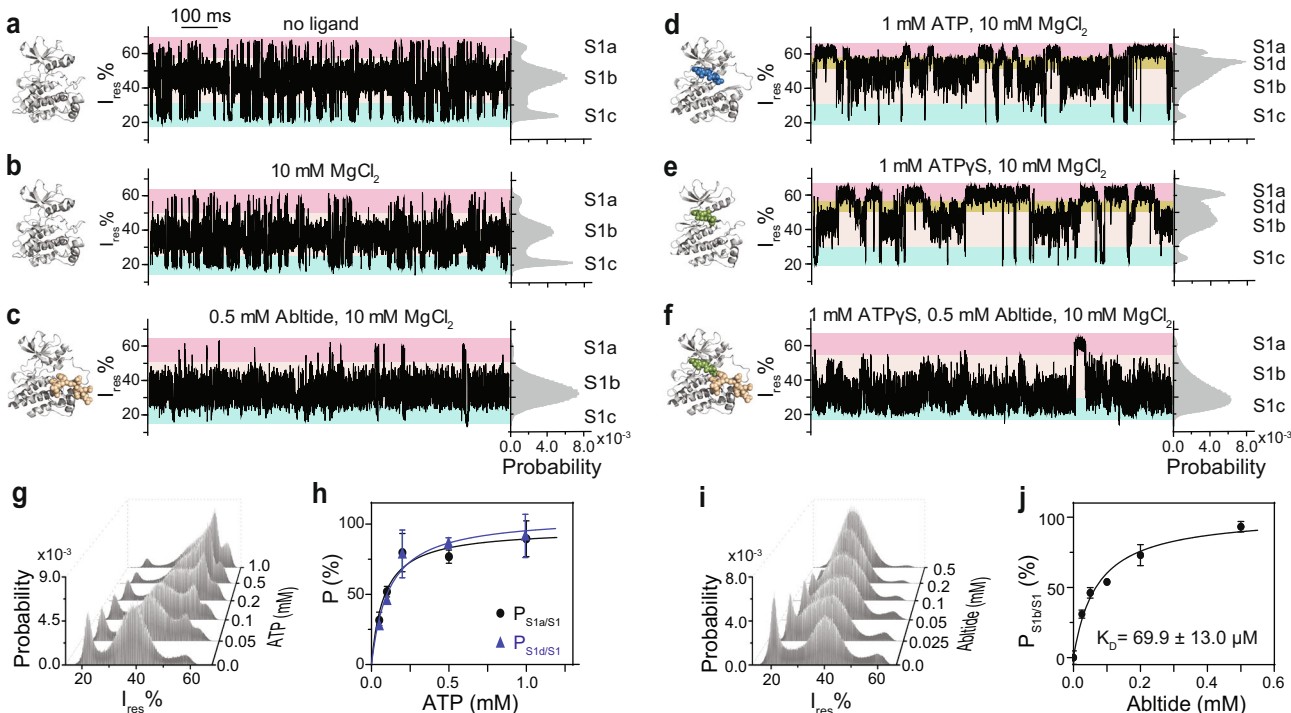

**Fig. 2 Interactions of substrates and analogue with $_{N4pos}$Abl. a–f** Structural models, representative traces and corresponding histograms of S1 of apo $_{N4pos}$Abl (**a**) and $_{N4pos}$Abl in the presence of 10 mM MgCl$_2$ (**b**), 0.5 mM Abltide and 10 mM MgCl$_2$ (**c**), 1 mM ATP and 10 mM MgCl$_2$ (**d**), 1 mM ATPγS and 10 mM MgCl$_2$ (**e**), 0.5 mM Abltide, 1 mM ATPγS and 10 mM MgCl$_2$ (**f**). Structure models of apo Abl kinase domain and Abl:ATPγS were produced from PDB 6XR6 and 2G2F respectively. Structural models of Abl:ATP, Abl:Abltide and Abl:ATPγS:Abltide complexes were generated from PDB 2G1T. Abl, ATP, ATPγS and Abltide are colored in gray, blue, green and wheat, respectively. S1 sub-states S1a, S1b, S1c and S1d were highlighted with pink, beige, cyan and yellow, respectively. **g** All points histograms of S1 states at various ATP concentrations. **h** Determination of K$_D$ for ATP binding in two binding modes, S1a (black) and S1d (blue). Data were represented as mean ± SD, $n = 3$ independent replicates. Source data are provided as a Source Data file. **i** All points histograms of S1 states at various Abltide concentrations. **j** Determination of K$_D$ for Abltide binding. Data were represented as mean ± SD, $n = 3$ independent replicates. Source data are provided as a Source Data file. All current traces were collected at −80 mV in 150 mM NaCl, 100 mM Tris-HCl, pH 7.5, at 22 °C.

increasing the relative occupancy $P_{S1b/S1}$ (defined as the population occupancy of S1b relative to S1) from $62.1 \pm 2.9\%$ to $93.6 \pm 0.3\%$ (Fig. 2c). Interestingly, ATP binding augmented a pre-existing S1a state by increasing $P_{S1a/S1}$ from $13.3 \pm 2.6\%$ to $19.3 \pm 2.2\%$. In addition, ATP also induced a new current state S1d with a $P_{S1d/S1}$ of $39.8 \pm 2.1\%$ (Fig. 2d), indicating the existence of two ATP binding modes. As expected, the binding of ATP to Abl kinase was Mg$^{2+}$-dependent as apo $_{N4pos}$Abl signal was readily recovered after adding EDTA (Supplementary Fig. 9). Similar to ATP, ATPγS also stabilized S1a state and induced a new state S1d, although showing a different population distribution: $P_{S1a/S1}$ of $30.0 \pm 2.0\%$ and $P_{S1d/S1}$ of $24.1 \pm 1.6\%$ (Fig. 2e). The ternary $_{N4pos}$Abl:ATP:Abltide complex was mimicked by an $_{N4pos}$Abl:ATPγS:Abltide state. The simultaneous binding of ATPγS and Abltide to $_{N4pos}$Abl stabilized a pre-existing state S1c from $25.0 \pm 3.2\%$ to $37.9 \pm 2.5\%$ (Fig. 2f). Interestingly no obvious signal change was detected in S2 state for all ligands (Supplementary Fig. 10). The distribution of binding state populations was concentration dependent allowing the determination of the dissociation constants (K$_D$) for each ligand (Fig. 2g–j). The K$_D$ was derived by fitting the relative population with the increasing ligand concentration as follows: $K_D^{ATP} = 81.6 \pm 17.9$ μM (S1a binding mode), $112.9 \pm 25.3$ μM (S1d binding mode), $K_D^{Abltide} = 69.9 \pm 13.0$ μM. Our $K_D^{ATP}$ values are consistent with the reported $69 \pm 4$ μM[45]. No K$_D$ value for this specific substrate peptide was determined previously.

Our data strongly indicate that S1 sub-states correspond to different ligand(s)-bound conformational states while S2 state

reflects a conformation incapable of binding substrates. Also, the observation that apo Abl kinase can sample different S1a, S1b, and S1c states which were selectively stabilized by different ligands, suggested a conformational selection mechanism for ligand recognition by Abl.

**S2 state is a non-ligand binding, lobe-open conformation.** Conformational switch between lobe-open and -closed states have been observed in many protein kinases and shown to play an essential role in catalysis and kinase activation[46–48]. Structures of active states loaded with substrates have been shown to adopt a lobe-closed conformation while residues forming the catalytic cleft are further away at the lobe-open[49,50]. Here we suspected that the S1 state might represent lobe-closed conformations allowing ligand/substrate binding while the S2 state might be a lobe-open conformation incapable of ligand binding. Studies on the extracellular signal-regulated kinase 2 (ERK2) suggested that the activation of ERK2 was promoted by substitution of the hinge domain residues with Gly to increase the flexibility of the hinge region that controls the N- and C-lobe domain movement[47,48]. To probe if S1/S2 states transition is related to the domain movement, we also introduced Gly into the hinge region (Phe$^{317}$-Leu$^{323}$) of Abl to generate two mutants ($_{N4pos}$Y320G and $_{N4pos}$T319G Y320G) with increased flexibility, as well as one mutant ($_{N4pos}$G321V) that intended to enhance rigidity (Fig. 3a). We then investigated how the hinge mutations affect the S1/S2 transition kinetics (Fig. 3b–f, Supplementary Table 5). Compared

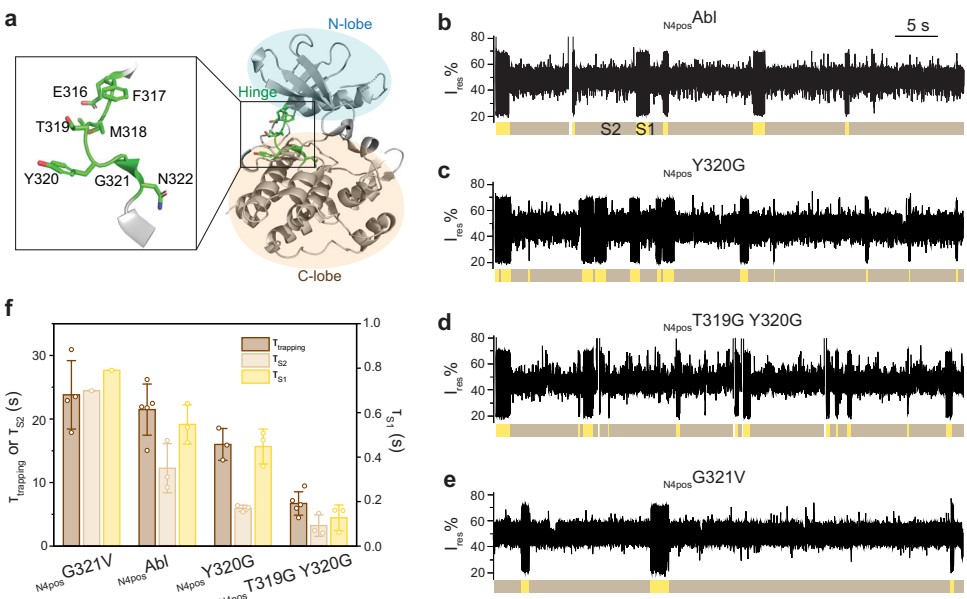

**Fig. 3 Effect of Abl kinase hinge mutations on the S1/S2 state transitions. a** Structure of Abl kinase domain (gray, PDB: 2HYY) with hinge region colored in green. N-lobe and C-lobe are indicated by the pale blue and beige circles respectively. A magnified image shows the detailed hinge structure with side chain of hinge residues (316-322) shown as stick. **b–e** Representative current traces of apo $_{N4pos}$Abl (**b**), $_{N4pos}$Y320G (**c**), $_{N4pos}$T319G Y320G (**d**) and $_{N4pos}$G321V (**e**) trapped with ClyA-AS nanopore tweezers. Signal patterns S1 and S2 are highlighted in yellow and brown, respectively. All current traces were collected at −80 mV in 150 mM NaCl, 100 mM Tris-HCl, pH 7.5, at 22 °C, with ~100 nM Abl kinase added to *cis* chamber. **f** Comparison of the dwell times of trapping and S1, S2 states of different Abl hinge mutants. Data were represented as mean ± SD, $n = 5$ independent replicates for $\tau_{trapping}$ of $_{N4pos}$Abl and $_{N4pos}$T319G Y320G, $n = 4$ independent replicates for $\tau_{trapping}$ of $_{N4pos}$G321V, $n = 3$ independent replicates for $\tau_{trapping}$ of $_{N4pos}$Y320G, $n = 3$ independent replicates for $\tau_{S1}$ and $\tau_{S2}$ of $_{N4pos}$Abl, $_{N4pos}$Y320G and $_{N4pos}$T319G Y320G. Of note all complete S1 or S2 from multiple experiments were combined to derive the $\tau_{S1}$ and $\tau_{S2}$ for $_{N4pos}$G321V. Consequently, we were not able to derive standard deviation for $\tau_{S1}$ and $\tau_{S2}$ for $_{N4pos}$G321V (see Methods for more details). Source data are provided as a Source Data file.

with $_{N4pos}$Abl, $\tau_{S1}$ of $_{N4pos}$Y320G decreased by ~18% and $\tau_{S2}$ decreased by ~2 folds, suggesting a faster S1/S2 transition. For $_{N4pos}$T319G Y320G, $\tau_{S1}$ decreased by ~2.3 folds compared to $_{N4pos}$Y320G and $\tau_{S2}$ by ~32%, suggesting more rapid S1/S2 transitions were further promoted with the double glycine mutations. In contrast, both $\tau_{S1}$ (~0.79 s) and $\tau_{S2}$ (~24.46 s) of $_{N4pos}$G321V showed an increase compared to $_{N4pos}$Abl, suggesting slower S1/S2 transitions. Interestingly, the trapping time decreased with the decreased hinge rigidity ($_{N4pos}$T319G Y320G < $_{N4pos}$Y320G < $_{N4pos}$Abl < $_{N4pos}$G321V), possibly due to the increased entropic cost of confining a more flexible protein within the nanopore (Fig. 3f). Together, all three mutants pointed out that the S1/S2 transition kinetics was sensitive to the hinge flexibility with a tendency that a more flexible hinge leads to faster transitions. Thus, our results strongly suggest that S1 and S2 states were induced by lobe motions with S1 likely resembling a lobe-closed conformation while S2 a lobe-open one.

**Imatinib stabilizes the S1c state.** Imatinib was shown to bind to ATP pocket by crystal structures (PDB: 2HYY[51], 1IEP[52], 3K5V[53]) with a long residence time ($1/k_{off}$) of ~23 min[54], making it a useful tool to lock Abl kinase in a specific conformational state for interrogation. Of note, imatinib was recently shown to bind Abl at the myristoyl pocket ($K_D$ = ~10 μM) at high imatinib concentration with a much shorter residence time of ~10 ms[55]. Thus, in the presence of 1 μM imatinib, we expected to mainly observe the Abl conformer loaded with imatinib at the ATP binding site. The current traces of imatinib bound $_{N4pos}$Abl no longer exhibited the hierarchical current states but a homogenous noisy current pattern showing a peak around the S1c state of apo $_{N4pos}$Abl (Fig. 4a–c). The dwell time of $_{N4pos}$Abl:imatinib complex was drastically increased to $103.9 \pm 3.5$ s ($N = 2$,

$n = 111$), which is 100 times longer than the $\tau_{S1c}$ and 5 times longer than the trapping time of apo $_{N4pos}$Abl. Our data indicated that imatinib binding stabilized a pre-existing conformational S1c state. Interestingly, the $_{N4pos}$Abl:ATPγS:Abltide ternary complex also appears to shift the equilibrium to S1c (Fig. 2f) despite the crystal structures showed that the activation loop of Abl:imatinib and Abl:ATP-peptide conjugate had closed and open conformations respectively (Supplementary Fig. 11a)[51,56]. Note $_{N4pos}$Abl:imatinib and $_{N4pos}$Abl:ATPγS:Abltide had distinct current patterns, both of which contained flickering events at ~200 μs scale suggesting the existence of fast motions of dynamic components within these two types of ligand-bound states (Supplementary Fig. 11b–c). Thus, we speculate that Abl:imatinib and Abl:ATPγS:Abltide complexes within ClyA may share a large similarity in their conformations while the crystallization condition captured a snapshot of different conformational transitions for each ligand-bound complex. In addition, this result further supported that the S1c belonged to a lobe-closed or ligand-bound conformation. We also determined the apparent $K_D$ of imatinib for Abl, defined as the imatinib concentration at which half of the Abl population were loaded with the inhibitor, to be $26.7 \pm 3.3$ nM using untagged Abl construct (Fig. 4d–e), consistent with previously known values, 1.5 ~ 47 nM [45,57,58].

**Mechanism underlying the catalytic inactivity of Abl G321V.** The G321V mutation abrogated the catalytic activity of Abl kinase by a mechanism unknown (Supplementary Fig. 2)[59]. Detailed examination of the current traces revealed that the $_{N4pos}$G321V mutant generated a significantly different S1 sub-state distribution from that of $_{N4pos}$Abl (Fig. 5a–b, Supplementary Table 3). The S1a of G321V became the dominating sub-state with an 80% occupancy of all S1 state compared with 18%

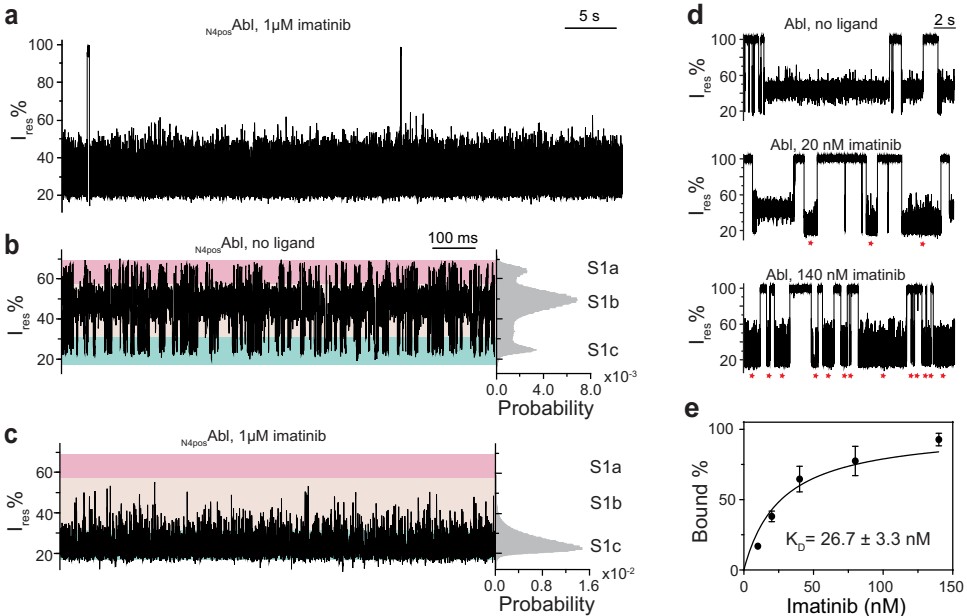

**Fig. 4 Interaction of imatinib with Abl kinase. a** A representative trace of ₙ₄ₚₒₛAbl trapped within ClyA-AS nanopore tweezers in the presence of 1 μM imatinib. **b, c** Representative zoomed-in traces and corresponding histograms of apo ₙ₄ₚₒₛAbl (**b**) and imatinib-bound ₙ₄ₚₒₛAbl (**c**) observed with ClyA-AS nanopore tweezers. S1 sub-states S1a, S1b and S1c were highlighted with pink, beige and cyan, respectively. The current traces were collected at −80 mV in the buffer 150 mM NaCl, 100 mM Tris-HCl, pH 7.5, at 22 °C, with ~100 nM ₙ₄ₚₒₛAbl added to *cis* chamber. **d** Representative traces of Abl titrated with different concentrations of imatinib (0, 20 nM, 140 nM). Red star indicates imatinib-bound Abl events. The current traces were collected at −90 mV in the buffer 150 mM NaCl, 100 mM Tris-HCl, pH 7.5, at 22 °C, with ~100 nM Abl and different amount of imatinib added to *cis*. **e** Determination of the apparent $K_D$ of imatinib by plotting the imatinib bound event population over imatinib concentration. Data were represented as mean ± SD, $n = 3$ independent replicates. Source data are provided as a Source Data file.

for ₙ₄ₚₒₛAbl, indicating that the G321V mutation not only altered the lobe-motions (S1/S2 transitions) but also affected the conformational distributions during the S1 lobe-closed state.

We next examined the ability of ₙ₄ₚₒₛG321V to interact with substrates. Strikingly, the addition of ATP (or ATPγS) and MgCl₂ did not trigger any change in the current trace suggesting that ₙ₄ₚₒₛG321V may be deficient in ATP and ATPγS binding (Fig. 5b–c, Supplementary Fig. 12). It was proposed that the ATP binding triggered the assembling of a conserved hydrophobic spine, C-spine between the N- and C-lobe interface is crucial for kinase activity[60–63]. The mutation G321V in ATP binding pocket was close to the residues constituting the C-spine (Fig. 5e). Because G321V exhibited an increased S1a sub-state, we speculate that G321V mutation may enhance hydrophobic interactions of the C-spine residues and promote the assembling of the C-spine to mimic an ATP-like binding state in the absence of ATP (Fig. 5a–b, Supplementary Table 3). In addition, the bulkier side chain of V321 could clash with the ribose group of ATP, making ATP geometrically unfit for the mutated ATP-binding pocket. We also tested the binding ability of ₙ₄ₚₒₛG321V to Abltide (Fig. 5b, d). Interestingly, the $K_D^{Abltide}$ of ₙ₄ₚₒₛG321V (257.3 ± 55.9 μM) decreased by ~3.7 folds compared with that of ₙ₄ₚₒₛAbl despite the lack of direct contact between G321V and Abltide, suggesting the hinge region allosterically regulates peptide binding (Fig. 5f–g). Together, these results strongly pointed out that the loss of catalytic activity of G321V mutant was due to its inability in ATP binding.

**The conformational energy landscapes of apo ₙ₄ₚₒₛAbl.** Our nanopore tweezers showed that the apo Abl kinases exhibit an ensemble of conformational states in solution. Quantification of the thermodynamic and kinetic parameters of those conformations enabled us to propose a comprehensive conformational

energy landscape model for ₙ₄ₚₒₛAbl (Fig. 6, Supplementary Table 6). There are two main basins in the energy landscapes of apo ₙ₄ₚₒₛAbl corresponding to two main conformational states, one higher-energy lobe-closed conformation (S1) and the other lower-energy lobe-open conformation (S2). The transition rate constant from S1 to S2 state or lobe open rate constant ($k_{S1→S2}$) was 1.86 ± 0.24 s⁻¹ and lobe close rate constant ($k_{S2→S1}$) was 0.09 ± 0.24 s⁻¹. The free energy difference between S1 and S2 state ($\Delta G_{S1,S2}$) was 7.57 ± 0.91 KJ mol⁻¹. Inside the basin S1, there are three sub-basins (a, b, c) separated by smaller energy barriers, corresponding to three conformational sub-states, S1a, S1b, and S1c, which resemble different substrate-binding states (Fig. 2). The transition rate constants of $k_{S1a/c→S1b}$ and $k_{S1b→S1a/c}$ was 154.1 and 97.9 s⁻¹ respectively. Notably, S1a and S1c share nearly equal free energy with a very small $\Delta G_{S1a, S1c}$ (0.42 ± 0.21 KJ mol⁻¹). The S1a and S2c states also show rapid transitions ($k_{S1a→S1c} = 974$ s⁻¹ and $k_{S1c→S1a} = 824$ s⁻¹).

In comparison, a recent NMR study performed at 10 °C showed that apo Abl kinase interconverted between three states linearly, G ↔ E1 ↔ E2[22]. The G state dominated the population by 88% while E1 and E2 had a population of 6% for each. Using the ¹³C-based chemical exchange saturation transfer (CEST), the exchange rate ($k_{ex}$) for the G ↔ E1 transition and E1 ↔ E2 transition were determined as 46.8 ± 4.3 s⁻¹ and 88.7 ± 13.5 s⁻¹, respectively[22]. Note, the lobe close and open motions revealed by our nanopore tweezers at room temperature have slow rates ($k_{S2→S1} = 0.09 ± 0.02$ s⁻¹, $k_{S1→S2} = 1.86 ± 0.24$ s⁻¹). At 10 °C, such motions would further slow down, which likely become invisible in the ¹³C-based CEST measurements that focused on the time scale between ~20 and 500 s⁻¹[64]. Another possibility is that Abl preferentially adopted a lobe closed conformation at the low temperature. Interestingly, the transition rates of three S1 sub-states in our nanopore measurements fell well in the detection

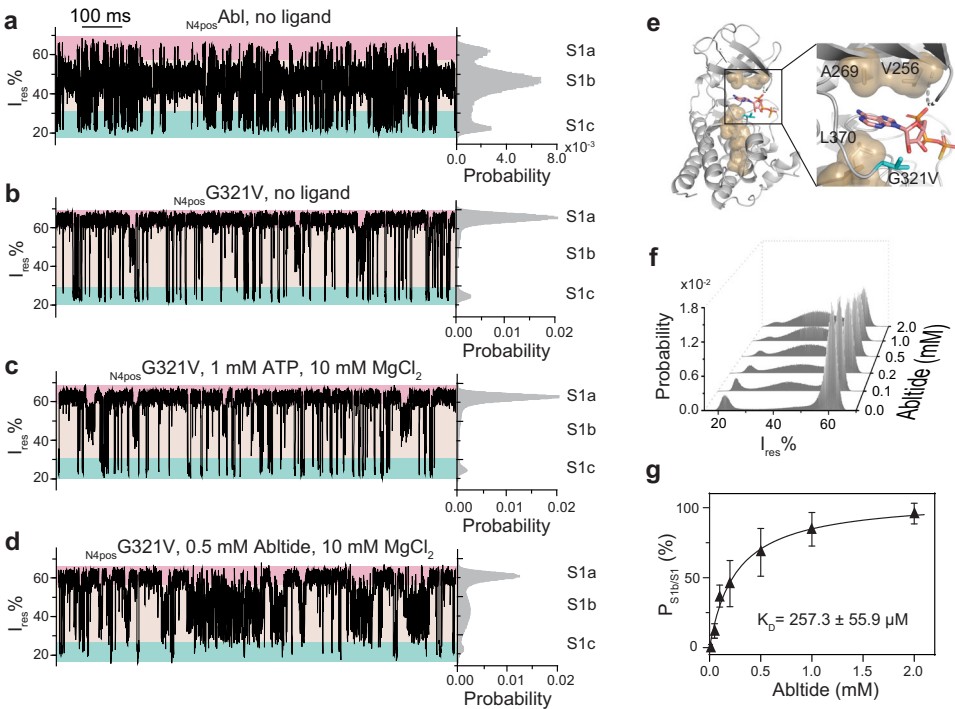

**Fig. 5 Trapping of catalytic inactive G321V mutant with ClyA-AS nanopore tweezers. a–d** Representative S1 state traces and corresponding histograms of apo $_{N4pos}$Abl (**a**), apo $_{N4pos}$G321V (**b**) and $_{N4pos}$G321V in the presence of 1 mM ATP and 10 mM $MgCl_2$ (**c**), 0.5 mM Abltide and 10 mM $MgCl_2$ (**d**). S1 sub-states S1a, S1b, and S1c were highlighted with pink, beige, and cyan, respectively. **e** Structure of G321V modeled from ATPγS-bound wildtype Abl kinase structure (PDB ID: 2G2F). ATPγS is shown as a stick with carbon atoms colored in pink. G321V mutation is shown as a stick and colored in cyan. Hydrophobic residues forming the C-spine are shown in the surface model and highlighted in sand color, V256, A269, and L370 are close to the ATP/ATPγS binding site. **f** Histogram of S1 state for Abltide titration to $_{N4pos}$G321V. **g** Determination of the binding affinity of Abltide to $_{N4pos}$G321V based on the S1 state histograms. Data were represented as mean ± SD, $n = 3$ independent replicates. Source data are provided as a Source Data file. The current traces were collected at −80 mV in buffer 150 mM NaCl, 100 mM Tris-HCl, pH 7.5, at 22 °C.

range of the NMR study. Therefore, we propose that the three sub-states S1a, S1c and S1b may originate from the local re-arrangement of the G-loop, A-loop, DFG, and the αC-helix.

Current signal fluctuations of the trapped protein could have two origins: (1) alteration of the protein-ClyA interface caused by the conformational changes of the protein; (2) metastable protein/nanopore interactions, e.g., protein moves between different sites of lumen or tumbling inside of nanopore forming multiple interaction interfaces. In the latter scenario, the current states induced by protein movement are functionally irrelevant. The formation of protein-ClyA interaction interface is controlled by a combination of factors including electrophoresis, electroosmosis, the shape, net charge, and charge distribution of the protein, the physical properties of nanopore lumen, and the steric hindrance. Multiple pieces of evidence supported that the observed current states were induced by kinase conformational changes that are functionally relevant. First, the binding of different substrates or substrate combinations stabilized unique pre-existing S1 sub-states of apo $_{N4pos}$Abl (S1a, S1b, S1c), suggesting that the S1 sub-states resembled the substrate-binding conformations. Note, substrate (MgATP or Abtide) binding not only altered the conformation of Abl, but also introduced additional negative or positive charges to holo $_{N4pos}$Abl. Should S1 substates originate from apo $_{n4pos}$Abl moving around of ClyA without undergoing conformational changes, one can only deduce that three holo $_{N4pos}$Abl states coincidentally stabilized each of the apo $_{N4pos}$Abl/ClyA binding interfaces despite of their distinct conformations and charge states of holo $_{N4pos}$Abl from those of apo $_{N4pos}$Abl. We believed such coincidence improbable. Second, the substitution of hinge residues of Abl kinase to non-charged amino acids (Y320G, G321V, and T319G Y320G) altered the population

and dwell time of S1 and S2 states, suggesting that both S1 and S2 were associated with conformational states that were regulated by the hinge motions of the kinase. Particularly, the faster S1/S2 transition rates of these mutants were tightly correlated with the increasing flexibility of the hinge. Third, after saturating the kinase with imatinib, the S1a, S1b, and S2 states were depleted and the Abl:imatinib complex exhibited current levels close to the S1c state, supporting that S1c represented a closed conformation revealed by crystal structures of Abl:imatinib complex. Together, our data strongly indicate that apo Abl kinase interconverted between two major conformations and three conformational sub-states during lobe-closed conformation. Note, previous ClyA nanopore tweezer studies have shown that the kinetics of ligand binding could be influenced by the applied potential and the confinement of nanopore imposed on trapped enzymes[31,65]. It is well known that an electric field can stimulate protein motions[66,67]. Given more than 25% of residues of the Abl kinase domain are charged, we anticipated that the electric field would have an impact on the structural dynamics of Abl within ClyA nanopore. Further ClyA nanopore tweezer analysis, in combine with molecule simulations will be carried out to explore how ligands, inhibitor molecules and drug-resistant mutations re-shape the conformational landscapes of Abl kinase, and to what extent the applied potential modulates the kinetics of these functional motions in the future.

Here we have demonstrated nanopore tweezers that can measure structural dynamics at a time scale (ms ~ 10 s). However, recordings at a broader time scale (100 μs ~ hr) while still maintain a high temporal resolution at 100 μs are easily achievable with commercial electrophysiological instruments. More importantly, nanopore devices could operate in high

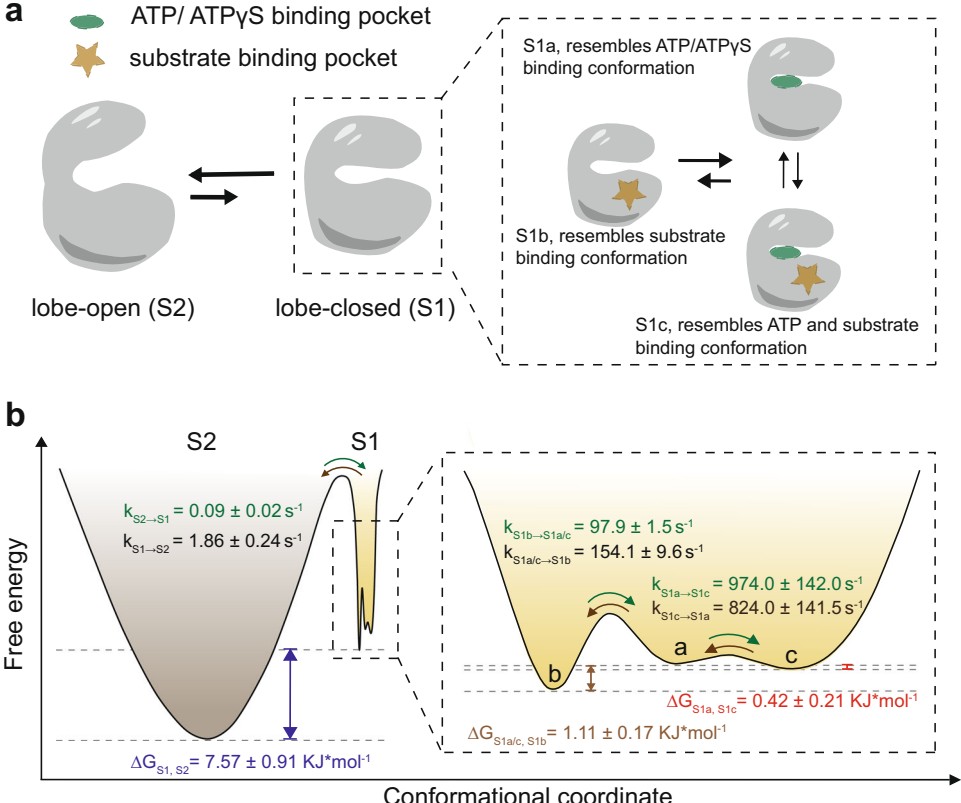

**Fig. 6 Conformational dynamics of apo Abl kinase domain. a** Models of apo Abl kinase domain conformational changes. Two major conformations (S1: lobe-closed, S2: lobe-open) and three S1 conformational sub-states (S1a, S1b, S1c) which resemble substrate or substrate combination binding states are shown. **b** Schematics of the conformational energy landscape of apo $_{N4pos}$Abl. S1 (yellow) and S2 (brown) indicate the two major conformations, lobe-closed and lobe-open, respectively. S1 conformational sub-states, S1a, S1b, S1c are indicated by a, b, c, respectively. State transition rate constant (k), and free energy difference (ΔG) between states are shown on the graphs.

throughput with massive multiplexing capabilities at low-cost[68]. Thus, we expect nanopore tweezers to be a paradigm shift that will expand the single-molecule protein structural dynamic studies from the realm of fundamental research in specialized laboratories toward more broad areas in drug discovery, and medical diagnostics, and precision medicine.

## Methods

**Mutagenesis, expression, and purification of Abl kinase.** Different Abl mutant plasmids were generated via PCR with mutagenesis primers listed in Supplementary Table 7. The Abl kinase containing plasmid pET_His10 TEV_Abl1 kinase domain (residue 229-512) (Addgene plasmid # 79727) and the YopH phosphatase containing plasmid pET_YopH (residues 164-468) (Addgene plasmid # 79749) were co-transformed to BL21(DE3)pLysS competent cells for protein expression[69]. Both plasmids were gifts from John Chodera & Nicholas Levinson & Markus Seeliger. Co-expression with the YopH phosphatase is to prevent potential autophosphorylation of Abl kinase. Cells were grown in 2 × YT media at a 37 °C shaker until OD$_{600nm}$ reached 0.8~1. Cells were then induced with 0.25 mM iso-propyl β-D-1-thiogalactopyranoside (IPTG) for 16 h at 16 °C. Afterwards, cells were harvested and resuspended in lysis buffer (50 mM Tris-HCl, 150 mM NaCl, 5% glycerol, pH 8) and lysed via sonicator (Misonix). The cells were pelleted by centrifugation at 20,000 × g for 20 min at 4 °C, and the supernatant was purified with Ni-NTA affinity chromatography. The proteins were washed with buffer containing 50 mM Tris-HCl, pH 8, 500 mM NaCl, 75 mM imidazole, 5% glycerol and eluted in buffer containing 50 mM Tris-HCl, pH 8, 500 mM NaCl, 200 mM imidazole, 5% glycerol. TEV protease was added to the elutions from Ni-NTA with a mole ratio of 1:15 (TEV:kinase) to remove the His-tag while dialyzed for 16 h at 4 °C in dialysis buffer (50 mM Tris-HCl, 100 mM NaCl, 5% glycerol, 1 mM DTT, pH 8). The dialyzed solution was then applied to an anion exchange column to remove TEV and phosphatase. The proteins were washed with buffer containing 50 mM Tris-HCl, pH 8, 100 mM NaCl, 5% glycerol, 1 mM DTT and eluted in buffer containing 50 mM Tris-HCl, pH 8, 200 mM NaCl, 5% glycerol, 1 mM DTT. The elute from anion exchange purification was concentrated by centricon with 10 000 Da cutoff and loaded to a size-exclusion column (HW5SS, GE Healthcare Life

Sciences) to remove potential aggregates with running buffer (50 mM Tris-HCl, 150 mM NaCl, 5% glycerol, 1 mM DTT, pH 8). All Abl kinase mutants were expressed and purified in a similar fashion as wild-type Abl. The purity of Abl proteins was examined by 12% sodium-dodecyl sulfate-polyacrylamide gel elec-trophoresis (SDS-PAGE) (Supplementary Fig. 13).

**Abl Kinase assay.** The activities of Abl proteins were measured using an enzyme-coupled assay[69], where the consumption of NADH (absorption at 340 nm) correlated to the phosphate transfer. Reactions were performed at 30 °C with 30 nM kinase, 10 mM MgCl$_2$, 100 mM Tris (pH 8.0), 2.2 mM ATP, 1 mM phosphoenolpyruvate, 0.6 mg/mL NADH, 75 U/mL pyruvate kinase, 105 U/mL lactate dehydrogenase, and 0.5 mM kinase substrate peptide Abltide (sequence: KKGEAIYAAPFA). Absorbance at 340 nm was monitored with a microtiter plate spectrophotometer (BioTek Synergy 2) for 30 min. The background kinase activity was determined in the reaction mix without the substrate peptide and subtracted from the experiments with the substrate peptide. To obtain the enzyme kinetics, initial phosphorylation rates at different substrate peptide concentrations (50~800 μM) were measured. Plots of the initial rate as a function of peptide substrate concentration were fit by nonlinear regression to obtain the Michaelis-Menten kinetic parameters.

**Preparation of ClyA-AS nanopore.** ClyA-AS containing plasmid pET3a-his5-ClyA-AS was transformed into BL21(DE3) E. coli competent cells for protein expression[70]. Cells were grown in LB media at 37 °C until OD$_{600}$ reached ~0.6. Then IPTG was added with a final concentration of 0.5 mM to induce the cells and incubated at 16 °C for ~16 h. The cells were harvested and resuspended in lysis buffer (50 mM Tris-HCl, 150 mM NaCl, 5% glycerol, pH 8) and lysed via sonicator (Misonix). The cells were pelleted by centrifugation at 20 000 g for 20 min at 4 °C. The supernatant was then purified with gravity Ni-NTA affinity columns equili-brated with a buffer containing 150 mM NaCl, 20 mM imidazole, 50 mM Tris-HCl, pH 8. Wash buffer (150 mM NaCl, 100 mM imidazole, 50 mM Tris-HCl, pH 8) was then applied to the column to wash off impurities. The ClyA-AS was eluted with elution buffer (150 mM NaCl, 200 mM imidazole, 50 mM Tris-HCl, pH 8). The elution from Ni-NTA columns was further purified by a size-exclusion column (HW5SS, GE Healthcare Life Sciences) with running buffer (150 mM NaCl, 50 mM Tris-HCl, pH 8) to remove the potential aggregates. The protein purity was

assessed using SDS-PAGE analyses (Supplementary Fig. 13). Proteins were frozen in liquid nitrogen and stored at −80 °C for future use. The ClyA-AS monomer was assembled into oligomer by incubating with 1% DDM at room temperature for 30 min and then stored in a 4 °C fridge for current recording experiments.

**Single-channel current recording and data analysis**. Single-channel current recording was performed at room temperature 22 °C and pClamp 10.7 was used for data acquisition[31]. Briefly, a ClyA-AS nanopore was inserted into a DPhPC planar lipid bilayer separating two chambers. Both chambers were filled with 300 µL buffer (150 mM NaCl, 100 mM Tris-HCl, pH 7.5). ClyA-AS was added to the *cis* chamber that was grounded and then spontaneously inserted into the bilayer. Abl kinase (~100 nM) was added to the same chamber. After applying a negative voltage bias across the bilayer, the current generated by the ions flow through the nanopore was monitored in voltage-clamp mode by an integrated patch clamp amplifier (Axo-patch 200B, Molecular Devices). The current signal was acquired by an analog-to-digital converter (Digidata 1440 A, Molecular Devices) at a sampling rate of 50 kHz after processing with a 4-pole lowpass Bessel filter at 2 kHz.

Clampit 11.1 was used for data analysis acquisition[31]. Residual current ($I_{res}$) was calculated from blocked pore current ($I_B$) and open pore current ($I_O$) with $I_{res}\% = 100\% \times I_B/I_O$. The current range of peaks in the histogram was used to define the S1 substates. Specifically, the Ires% range of N4posAbl S1 substates under different conditions at −80 mV are defined as S1a (49 ~ 64%), S1b (28 ~ 49%), S1c (15 ~ 28%), S1d (40 ~ 56%). At apo state, the S1a, S1b, and S1c were clearly distinguishable by Ires% while the nucleotide binding generated a new S1d state whose current distributions largely overlapped with S1a and S1b. Dwell times of trapping events and current states were determined by using single-channel search in Clampfit 11.1. All dwell times from one nanopore experiment were binned and fitted to a single-exponential function to derive the τ. The average dwell times and the standard deviation were determined from at least three independent nanopore measurements. Of note in calculating $\tau_{S1}$ and $\tau_{S2}$, the S1 or S2 states at the beginning of a trapping event were discarded as their true dwell times were interrupted by Abl entering and exiting the ClyA-AS nanopore (see Supplementary Fig. 5). However, due to the super slow transition between S1 and S2 in $_{N4pos}$G321V, we were not able to collect enough complete S1 states for a good single-exponential fitting from one experiment. Therefore, all complete S1 or S2 from multiple experiments were combined to derive the $\tau_{S1}$ and $\tau_{S2}$ for $_{N4pos}$G321V. Consequently, we were not able to derive standard deviation for $\tau_{S1}$ and $\tau_{S2}$ for $_{N4pos}$G321V. State transition rate constant ($k$) was calculated with equation: $k = 1/\tau$. For example, the transition rate constant from S1 to S2 state, $k_{S1 \to S2} = 1/\tau_{S1}$. State population ($P$) was calculated by state dwell times, for example, $P_{S1} = \tau_{S1}/(\tau_{S1} + \tau_{S2})$. Relative state occupancy in the presence of different ligands was calculated by peak area in histograms. The free energy difference between two states was calculated with $\Delta G = -RT\ln K_{eq}$. For example, $\Delta G_{S1, S2} = -RT\ln K_{eq} = -RT\ln(\tau_{S1}/\tau_{S2})$. The dissociation constant ($K_D$) of the ATP or Abltide or imatinib from Abl kinase was determined from titration experiments. Plots of bound state occupancy as a function of titrate (ATP or Abltide) concentration were fit by nonlinear regression with single site binding model to obtain $K_D$. The $K_D$ of imatinib to Abl was determined by plotting the Bound % as a function of imatinib concentration and fitted by nonlinear regression with single-site binding model. Bound % $= 100\% \times \frac{\text{number of imatinib bound event}}{\text{number of total event}}$.

**Reporting summary**. Further information on research design is available in the Nature Research Reporting Summary linked to this article.

## Data availability

The datasets generated during and/or analyzed during the current study are available from the corresponding author on request. The source data underlying Figs. 1f, 2h, j, 3f, 4e, 5g and Supplementary Figs. 2 and 3, Supplementary Tables 1–6 are provided in the Source data file. PDB entries 2G1T [https://doi.org/10.2210/pdb2G1T/pdb], 6XR6 [https://doi.org/10.2210/pdb6XR6/pdb], 2G2F [https://doi.org/10.2210/pdb2G2F/pdb] and 2HYY [https://doi.org/10.2210/pdb2HYY/pdb] were downloaded from the Protein Data Bank and used in this article for molecular visualizations. Source data are provided with this paper.

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

## Acknowledgements

Plasmid pET3a-his5-ClyA-AS was a generous gift from Dr. Giovanni Maglia. We appreciate all valuable discussions with Drs. Margaret Stratton, Bach Pham, Xin Li, Jiale Du, Kirandeep Deol, Meizhe Wang, Patanachai 'Kong' Limpikirati, and Senyao He. Research in the Chen lab was supported by grants R01GM115442 (M.C.) and R01AI156187 (M.C.) from the US National Institutes of Health.

## Author contributions

F.L. designed and did the experiments, analyzed the data, and wrote the manuscript. M.A.F. analyzed the current recording data and participated in writing the manuscript. K.M.G. and R.P. performed the current recording experiments and analyzed the data. M.C. designed the experiments, wrote the manuscript, and supervised the project.

## Competing interests

The authors declare no competing interests.
