## [Peer Review File · Nature Communications]

Fanjun and coworkers investigated the conformational flexibility of Abl kinase using a biological nanopore. They found that inside the nanopore, the apo enzyme fluctuates between different current states. The addition of different ligands or substrates shifted the current blockade to different minima, allowing to assigning different conformation (identified by structural studies) to the observed current levels. Protein mutations confirmed the link between the conformational flexibility of the enzyme with catalysis.

This is an important work and step towards the use of biological nanopores as single molecule sensors for protein dynamics. Indeed, kinases are important protein systems in which enzyme dynamics play important roles in the catalytic mechanism. This work brought nanopore a step forward as single-molecule sensors for protein dynamics.

However, I think the authors should perform extra experiments/analysis to prove that the different current level are indeed representing dynamics states of the protein. Since the current levels can have different molecular interpretations, I find important this link is established beyond any reasonable doubt.

The applied potential can have an effect on the protein stability / conformational exchange. As the authors themselves invoke, the different current levels could reflect the rattling / tumbling of the protein within the nanopore rather than intrinsic structural fluctuation of the enzyme. A voltage dependent analysis of the blockades should be performed. If the kinetics do not change significantly with the potential, then they can be more safely considered as intrinsic enzyme kinetics. If they are voltage dependent, extrapolations to zero applied potentials should reveal if they are due to intrinsic kinetics or are more likely to be motions of the protein inside the nanopore. Hence, The authors should carefully analyze S1 and S2, and S1a,b,c kinetics for all mutants at different trapping potentials (when possible). If the kinetics change with the potential this is could be an indication of trapping kinetics rather than intrinsic kinetics. I

The author assigned S1a, S1C and S1b to the states G, E1 and E2 states observed by the recent NMR study (DOI: 10.1126/science.abc2754). This correlation is very important, because it can provides a convincing argument to link the current levels with structure. Overall, throughout the paper, however, the link between the states should be made clearer. The authors should establish this connection early on and refer to the papers that have characterized these states. Then they should aim at establish if S1a-c are indeed the states identified by NMR. In particular a direct comparison between S1a, S1C and S1b should be done for the WT and a few mutants (it is actually not really if the authors measure S1a, S1C and S1b for the mutants). Kinetics measurements between S1a-c appear measurable and should be affected by mutations in a similar manner as described in the NMR paper. S1a, S1C and S1b for G321V was indeed reported but no rates were reported and as far as I could see this mutant was not measured by NMR. In general, the modifications that induce the larger difference in the NMR study should be tested quantitatively by nanopore currents, and directly compared.

Strangly, S1 and S2 appear different in the different constructs (see below). In the ABIC4pos construct, S2 appears less represented (please add the color-code analysis of the two states in Figure S2). If true, this suggests that the two states are reflecting the movement of the protein inside the nanopore rather than the intrinsic dynamics of the trapped enzyme.

Fig 1

Fig S2

The authors should add longer traces (e.g. 20 sec) for the data showed in Figure 2a-f.

Figure 3f. How can the trapping time be shorter than the dwell time of S2?

Page 4. Please indicate how I_{res} was calculated (e.g by indicating the open pore current and blocked pore current).

Page 5.

Figure 2d. Is S1d an independent state or simply S1b but shifted? Can this be proven?

Is the ternary complex sampled using ATP or gamma-ATP?

The authors indicate a KD. However, it is not really clear what is the meaning of the KD. Is there any dependence on the applied potential (can be tested) or confinement inside the nanopore (should be briefly discussed)?

There is no evidence that the recognition is either conformational selection or induced fit. The authors should either elaborate on their interpretation or remove this sentence.

Page 9-10. How were the ΔG calculated?

Page 10. The authors correlate S1 and S2 with the altered dynamics of the mutated proteins. However, the authors argued earlier that the dynamics **important** for functions (NMR study) are those involving the S1a-c current levels rather S1 and S2. This is making the argument confusing. It appears more logic to compare the S1a-c current levels between mutants and related them to the NMR data (also mutants were tested there).

Reviewer #2 (Remarks to the Author):

This is an interesting work that uses tweezers to look at the conformational ensemble of Abl kinase at the single-molecule level. Although by itself this approach cannot provide structural information, combined with the recent atomic insight into the various conformational states of Abl by NMR, it can be very informative. The findings are of interest, especially considering how challenging going it is to extract information on a protein that is rapidly interconverting between conformational states. If this approach can be further developed and refined it could be a very useful complement to NMR.

In the experiment with imatinib, the authors should bring to the attention of the reader that the drug can also bind allosterically to Abl (PMID 34774565). This is not an issue for the current work since the authors work at low concentration, but it is something that the authors need to discuss.

“Mapping the conformational energy landscape of Abl kinase using ClyA nanopore tweezer” by Fanjun Li et al. described an approach using ClyA nanopore to investigate the conformational dynamics of Abelson (Abl) kinase. Two major conformational states of Abl (S1 and S2) and three sub-states of S1 have been characterized (S1a, S1b, and S1c). Moreover, the authors investigated the interaction of ligands, substrates with $N_{4\text{pos}}\text{Abl}$ (Abl with a positively charged peptide tail at its N-terminus), suggesting that sub-states of S1 reflect the ligand/substrate binding conformations while S2 state is non-ligand conformation. Several mutants have been designed to check the S1/S2 transition kinetics and their interaction with substrates. Overall, this is a very interesting manuscript reporting an important development in the nanopore field. The results obtained in this study have a potential impact on applying nanopores to study the conformational dynamics of proteins. However, there are some questions that need to be addressed before publication.

Major:

1. In the introduction part (Page 2), the authors stated “...but it remains challenging to obtain state transition kinetics owing to the large system size and long timescales associated with kinase conformational dynamics.” Could the authors explain why the large system size and long timescales conformational dynamics make it challenging to obtain state transition kinetics? Could the authors comment/introduce the currently available methods/studies that investigated the conformational dynamics of Abelson kinase?

2. The authors stated “...(ClyA) nanopore variants to reveal their ligand bound and catalytic states. Notably, four different ionic states of dihydrofolate reductase (DHFR) were detected³⁴ and three anomeric maltose-bound states of maltose binding protein (MBP) were resolved by ClyA nanopores...” Could the authors comment on what’s the difference or highlight between their manuscript and the papers that have been cited here since these studies already have proved that ClyA nanopores are able to detect different ligand bindings, catalytic states and three anomeric binding states of target proteins?

3. Page 3, Fig. 1 caption “...The current traces were collected at -80 mV in 150 mM NaCl, 100 mM Tris-HCl, pH 7.5...” Could the author explain why -80 mV has been chosen for collecting the data? I would assume the application of voltage may have some effect on the trapping time of protein inside the nanopore and therefore possibly affect the kinetics, especially the authors introduced a positively charged tag in the protein here. Could the authors show the relationship between trapping time and different voltages of Abl, $\text{Abl}_{\text{C4pos}}$ and $N_{4\text{pos}}\text{Abl}$, for instance, -60, -70, -80, -90, -100mV, respectively.

4. Page 4, “...two Abl constructs with a positively charged peptide tail at its N-terminus, termed $N_{4\text{pos}}\text{Abl}$, or to its C-terminus, named $\text{Abl}_{\text{C4pos}}$. Both $\text{Abl}_{\text{C4pos}}$ and $N_{4\text{pos}}\text{Ab}$; showed longer τ_{trapping} of 1.52 ± 0.13 s (N=3) and 20.72 ± 2.83 s (N=3), respectively...” Could the authors show what’s the sequence of positively charged peptide tail? Although the authors described “...plasmid pET_His10 TEV” in the method part, it’s not clear; the authors stated the data from 3 different pores but didn’t mention how many events have been collected from each pore to get statistic results; It’s also interesting that $\text{Abl}_{\text{C4pos}}$ and $N_{4\text{pos}}\text{Abl}$ show a very different τ_{trapping} , $N_{4\text{pos}}\text{Ab}$ is

almost 200 times longer than Abl_{C4pos}, could the authors comment about the possible reasons of such big difference? And why Abl_{C4pos} trapping time is not long enough to study the conformational energy landscape of Abl.

5. Page 4, the authors stated "...In current recording experiments, N_{4pos}Abl induced similar current signals as Abl (Fig. 1c-e, Supplementary Fig. 3)." However, why there is no data for Abl in Supplementary Table 2? According to Fig. 1c, Abl did show S1 and S2 states, please add it.

6. The Michaelis-Menten kinetics analysis of Abl_{C4pos} is missing, please add it in Supplementary Table 1 and Supplementary Figure 1.

7. Page 6, "...mutations were introduced into the hinge region (Phe³¹⁷-Leu³²³) of Abl¹¹ to generate two mutants (N_{4pos}Y320G and N_{4pos}T319G Y320G) with increased flexibility, and one mutant (N_{4pos}G321V) with enhanced rigidity". Could the authors refer to some references or use some structure data to demonstrate that mutated proteins here (N_{4pos}Y320G and N_{4pos}T319G Y320G) do have increased flexibility while N_{4pos}G321V shows an enhanced rigidity. Ref.11 didn't study these mutants.

8. If I'm not mistaken, the trace shown in Fig 2a, Fig. 4b and Fig. 5a are exactly the same. It will be just nice to see another trace of N_{4pos}Abl trapped within ClyA.

9. Page 10, "...while still maintain a high temporal resolution at 100 μs", could the authors explain why the resolution is 100 μs according to their description in method (Page 13) "...at a sampling rate of 50 kHz after processing with a 4-pole lowpass Bessel filter at 2 kHz".

10. Page 13, data analysis, how did the authors process the data? Clampfit or customized script? How did the author define peak S1a, b, c and d? For instance, the histogram shown in Fig .1c is actually in between S1b and S1c. Fig. 1d-e, how the authors assign the overlap part between S1d and S1b?

Minor:

1. Page 2, "...to characterize the dynamics A-loop", maybe "to characterize the dynamics of A-loop" is better?

2. Page 4, "...Both Abl_{C4pos} and N_{4pos}Ab; showed longer..." contains an apparent typo

3. Page 13, "State population (P) was calculated either by state dwell times, for example, PS1= τS1/(τS1 + τS2). Relative state occupancy in the presence of different ligands was calculated by peak area in histograms..." contains an uncompleted sentence and a typo.

Reply to Reviewers

We thank the reviewers for their careful inspection and critical assessment of our manuscript. In the revised manuscript, we have addressed the issues raised by the reviewers, with major changes highlighted in green. Our itemized responses are listed below.

Reviewer 1:

Fanjun and coworkers investigated the conformational flexibility of Abl kinase using a biological nanopore. They found that inside the nanopore, the apo enzyme fluctuates between different current states. The addition of different ligands or substrates shifted the current blockade to different minima, allowing to assigning different conformation (identified by structural studies) to the observed current levels. Protein mutations confirmed the link between the conformational flexibility of the enzyme with catalysis.

This is an important work and step towards the use of biological nanopores as single molecule sensors for protein dynamics. Indeed, kinases are important protein systems in which enzyme dynamics play important roles in the catalytic mechanism. This work brought nanopore a step forward as single-molecule sensors for protein dynamics.

Reply: We thank the reviewer for the positive recommendation.

However, I think the authors should perform extra experiments/analysis to prove that the different current level are indeed representing dynamics states of the protein. Since the current levels can have different molecular interpretations, I find important this link is established beyond any reasonable doubt.

1. The applied potential can have an effect on the protein stability / conformational exchange. As the authors themselves invoke, the different current levels could reflect the rattling/tumbling of the protein within the nanopore rather than intrinsic structural fluctuation of the enzyme. A voltage dependent analysis of the blockades should be performed. If the kinetics do not change significantly with the potential, then they can be more safely considered as intrinsic enzyme kinetics. If they are voltage dependent, extrapolations to zero applied potentials should reveal if they are due to intrinsic kinetics or are more likely to be motions of the protein inside the nanopore. Hence, The authors should carefully analyze S1 and S2, and S1a,b,c kinetics for all mutants at different trapping potentials (when possible). If the kinetics change with the potential this is could be an indication of trapping kinetics rather than intrinsic kinetics.

Reply: Thank you for the suggestion.

We performed voltage-dependent experiments with Abl proteins. Unfortunately, we were not able to extrapolate the voltage effect on the kinetics with the current protein design. We noticed that all the trapping events mostly started with the S1 signal and ended with the S2 signal (See **Supplementary Figure 5**), indicating that the Abl protein entered the nanopore in the S1 state and escaped from ClyA while in the S2 state (The reason for such phenomenon is to be investigated). Because of that, we believe the duration times of the initial S1 and the last S2 event from a trapping trace were truncated. Many short Abl-trapped events contained only one S1-S2 transition cycle. Although they were included in the calculation of the average trapping time, these S1 and S2 events were excluded from the calculation of τ_{S1} and τ_{S2} . For calculating τ_{S1} and τ_{S2} , we only used the S1 and S2 events in the middle of long trapping traces that contained multiple turnovers of S1 and S2 states with the first S1 and the last S2 events discarded (See **Supplementary Figure 5**).

The table below showed that at -80 mV, N_{4pos} Abl had the longest dwell time. Under this condition, multiple S1/S2 transition cycles took place within one trapping event, allowing us to accurately

calculate the dwell times (τ_{S1} and τ_{S2}) of S1 and S2 state from those completed S1, S2 events. At voltages below and above -80mV, the dwell time of $N_{4\text{pos}}\text{Abl}$ decreased by more than three times. Under these voltages, most events contained only one S1/S2 transition, i.e., partial (truncated) S1 and S2 events due to $N_{4\text{pos}}\text{Abl}$ entry and exit (see table below). Because of the large systematic error in the calculation of τ_{S1} and τ_{S2} at voltages below and above -80 mV, it is not meaningful to directly compare the kinetics of S1/S2 transitions of these voltages. Nevertheless, the data indicates the voltage did not affect the dwell time of S1 (τ_{S1}) significantly. The 50% decreased τ_{S1} at -70 mV, -90mV, and -100 mV was primarily due to the truncation of events. Should the voltage increase or reduce τ_{S1} by order of magnitude to ~2s or ~20 ms, the protein trapping time (>5 s) could allow the analysis to accurately reveal such changes. Our data also indicates that the voltages did not significantly decrease the dwell time of S2 (τ_{S2}) as the reduction of τ_{S2} at -70 mV, -90mV, and -100 mV was proportional to their reduced trapping times. Due to the shortened trapping time, whether a voltage would significantly increase the τ_{S2} cannot be predicted by the data obtained from the voltage-dependent experiments with the current protein/nanopore system.

For three sub-states S1a, S1b and S1c, the τ_{S1c} and τ_{S1ac} exhibited voltage dependence while the τ_{S1a} and τ_{S1b} did not. Here we respectfully disagree with the reviewer's notion that voltage-dependence of kinetics is a benchmark for the trapping kinetics (functionally irrelevant) instead of protein's intrinsic kinetics (functional-relevant). Because almost all proteins contain charged amino acids, the intrinsic motions of proteins should be inevitably affected by the electric field. External electric field-induced protein conformational changes have long been proposed and studied, both empirically and *in silico* (<https://doi.org/10.1002/bip.1976.360150904>; DOI: [10.1038/nature20571](https://doi.org/10.1038/nature20571); DOI: [10.1371/journal.pone.0221685](https://doi.org/10.1371/journal.pone.0221685)). However, to what extent the electric field can alter the different types of motions within a protein is dependent on the strength of the electric field and the sensitivity of the protein segments to the electric field. Using the motions of voltage-dependent Na^+ channels as an example: the transmembrane helix 4 (TM4) containing arginine residues is sensitive to the membrane potential while the selection filter component of the channel remains a rigid structure during the membrane polarization and depolarization processes. Here the observation that voltages changed the dwell time of some states of Abl kinase, but not others, could also be due to the state transitions were associated with motions of protein segments with different amino acid compositions.

V (mV)	τ_{trapping} (s)	τ_{S1} (s)	τ_{S2} (s)	τ_{S1a} (ms)	τ_{S1c} (ms)	τ_{S1b} (ms)	τ_{S1ac} (ms)
-60	0.14 ± 0.02	Too short trapping time, didn't analyze					
-70	4.93 ± 1.53	0.24 ± 0.03*	5.84 ± 2.15*	0.92 ± 0.02	0.71 ± 0.03	8.99 ± 0.24	4.93 ± 0.31
-80	21.48 ± 3.60	0.59 ± 0.07	13.28 ± 5.10	1.05 ± 0.15	1.26 ± 0.24	10.22 ± 0.16	6.52 ± 0.42
-90	6.86 ± 1.36	0.33 ± 0.09*	6.19 ± 0.56*	0.58 ± 0.03	2.30 ± 0.35	8.58 ± 1.04	12.90 ± 0.71
-100	6.55 ± 1.92	0.27 ± 0.07*	*5.54 ± 1.46*	0.55 ± 0.03	3.70 ± 0.82	9.45 ± 0.49	25.09 ± 5.1

*partially truncated events were also included for the calculation of the average dwell time.

Because the intricate voltage effect on the protein motions, understanding the voltage effect on the Abl kinetics at the molecule level requires re-design ClyA to increase the trapping time at a broad range of voltages, and create Abl mutants to assess the sensitivity of different part of proteins to electric field. We think this work is beyond the scope of the current manuscript. In the future, we will carry out a systematic analysis to investigate the voltage and nanopore confinement effect in detail.

In the revised manuscript, to emphasize that the Abl kinetics revealed by ClyA nanopore tweezers in the present study could be influenced by the voltage potential, we included a section in the Discussion (**Page 12**) as follows:

“Note, previous ClyA nanopore tweezer studies have shown that the kinetics of ligand binding could be influenced by the applied potential and the confinement of nanopore imposed on trapped enzymes. It is well known that an electric field can stimulate protein motions. Given more than 25% of residues of the Abl kinase domain are charged, we anticipated that the electric field would have an impact on the structural dynamics of Abl within ClyA nanopore. Further ClyA nanopore tweezer analysis, in combine with molecule simulations will be carried out to explore how ligands, inhibitor molecules and drug-resistant mutations re-shape the conformational landscapes of Abl kinase, and to what extent the applied potential modulates the kinetics of these functional motions in the future.”

2. The author assigned S1a, S1C and S1b to the states G, E1 and E2 states observed by the recent NMR study (DOI: 10.1126/science.abc2754). This correlation is very important, because it can provides a convincing argument to link the current levels with structure. Overall, throughout the paper, however, the link between the states should be made clearer. The authors should establish this connection early on and refer to the papers that have characterized these states. Then they should aim at establish if S1a-c are indeed the states identified by NMR. In particular a direct comparison between S1a, S1C and S1b should be done for the WT and a few mutants (it is actually not really if the authors measure S1a, S1C and S1b for the mutants). Kinetics measurements between S1a-c appear measurable and should be affected by mutations in a similar manner as described in the NMR paper. S1a, S1C and S1b for G321V was indeed reported but no rates were reported and as far as I could see this mutant was not measured by NMR. In general, the modifications that induce the larger difference in the NMR study should be tested quantitatively by nanopore currents, and directly compared.

Reply: In this manuscript, we did not intend to assign the observed S1 substates with the G, E1 and E2 states observed by the recent NMR study. Direct correlation of S1 substates with NMR states would not be possible as the energy landscape revealed by nanopore measurements was entirely different from the one in NMR study (See the table below). For example, we observed the slow S1/S2 transitions, which were likely induced by the domain movement and motion. However, the domain motions were not studied by the NMR. The dominate state of Abl revealed by ClyA nanopore tweezers was a lobe-open S2 while the NMR identified an active state (G) which differed from other states only in the distinct arrangements of A-loop, DFG and the α C-helix. It is noteworthy that the two studies were performed at different temperatures, 22°C for nanopore tweezers *versus* 10°C for NMR. It is possible that Abl was fixed at a lobe-closed conformation at the low temperature at which only conformation transitions with a low-energy barrier were allowed to occur. A recent work studying the temperature effect on an ion channel (DOI: [10.7554/eLife.59055](https://doi.org/10.7554/eLife.59055)) demonstrated that the channel open and close kinetics induced by Ca²⁺ binding exhibited very different profile at 21°C and 37°C. Also, a study about a psychrophilic adenylate kinase (<https://doi.org/10.1063/1.5089707>) showed that an I26T mutation significantly reduced catalytic activities at 55 °C, whereas the mutation showed no effect on the kinase activity at 45 °C. Thus, a mutation to Abl may increase the energy barrier between two states, the transitions of which were prohibited at 10°C while allowed at 22°C. In this case, the effect of the mutation measured at the two temperatures could be different. Therefore, we respectively disagree with the reviewer that the effects of mutations on the kinetics and the population distribution of conformational states observed by NMR and nanopore tweezers should be quantitatively comparable.

Perhaps one way to compare the two approaches is to perform the experiments at the same temperature. Unfortunately, our experimental set-up cannot operate at 10°C while the NMR studies

cannot achieve good resolution for different states at room temperature (DOI: 10.1126/science.abc2754). New development of the current recording instrument in the future is expected to broaden the utility of the nanopore tweezer approach.

To avoid the confusion, we revised the text (**page 11**) as “Therefore, we propose that the three sub-states S1a, S1c and S1b may originate from the local re-arrangement of the G-loop, A-loop, DFG and the α -helix.”

Dynamics of wild-type Abl kinase domain	Nanopore, 22°C				NMR, 10°C		
	S1, lobe closed state			S2, lobe open state	G	E1	E2
	S1a	S1b	S1c				
Transition pathway	S1b \leftrightarrow S1a or S1c; S1a \leftrightarrow S1c			S1 \leftrightarrow S2	G \leftrightarrow E1 \leftrightarrow E2		
Function	ATP, ATPyS binding	Abltide binding; dasatinib binding (data not shown)	ATPyS-Abltide complex binding; imatinib binding	not capable of ligand binding	active state	inactive state, PD173955 binding	inactive state, imatinib binding
Population	0.8 \pm 0.1 %	2.8 \pm 0.9 %	1.0 \pm 0.1 %	94.5 \pm 1.5 %	88%	6%	6%
Transition rate	$k_{S1a/c \rightarrow S1b} = 154.1 \pm 9.6 \text{ s}^{-1}$ $k_{S1b \rightarrow S1a/c} = 97.9 \pm 1.5 \text{ s}^{-1}$ $k_{S1a \rightarrow S1c} = 974.0 \pm 142.0 \text{ s}^{-1}$ $k_{S1c \rightarrow S1a} = 824.0 \pm 141.5 \text{ s}^{-1}$			$k_{S1 \rightarrow S2} = 1.86 \pm 0.24 \text{ s}^{-1}$ $k_{S2 \rightarrow S1} = 0.09 \pm 0.02 \text{ s}^{-1}$	exchange rate (k_{ex}) for the G \leftrightarrow E1 transition and E1 \leftrightarrow E2 transition were determined as $46.8 \pm 4.3 \text{ s}^{-1}$ and $88.7 \pm 13.5 \text{ s}^{-1}$, respectively.		

3.Strangly, S1 and S2 appear different in the different constructs (see below). In the ABIC4pos construct, S2 appears less represented (please add the color-code analysis of the two states in Figure S2). If true, this suggests that the two states are reflecting the movement of the protein inside the nanopore rather than the intrinsic dynamics of the trapped enzyme.

Reply: We’ve added the color-code analysis as shown in **Supplementary Fig. 6**.

In this Figure, the S2 state appears less represented in Abl_{C4pos}. But it should be noticed that the average trapping time of Abl_{C4pos} ($1.38 \pm 0.14 \text{ s}$) is around 15 times shorter than the trapping time of N4posAbl ($21.48 \pm 3.60 \text{ s}$) and even eight times shorter than the S2 duration time of N4posAbl ($12.25 \pm 3.16 \text{ s}$). Because of the short trapping time for Abl_{C4pos}, the duration of S2 states of Abl_{C4pos} was much shorter than its “real” duration. Thus, one cannot compare the duration times of S1 and S2 states of the two constructs.

In terms of the S1/S2 transition may reflect the movement of protein inside of nanopore. We think this is highly unlikely. Assuming S1/S2 states were caused by the movement of Abl, then S1 and S2 reflected two interaction interfaces between nanopore and Abl at the ground state. Under this assumption, the conformation of Abl remained unchanged during the S1/S2 transition with S2 being a more favorable/stable interaction state (95% population). However, we observed that mutations to the hinge, Y320G and T319G/Y320G mutants accelerate the S1/S2 transition while G321V retarded the S1/S2 transition. The changes in the transition kinetics would indicate mutations Y320G and

T319G/Y320G weakened the S1 and S2 interaction interface and G321 strengthened the S1 and S2 interaction interface.

Since all three mutations did not alter the charges of Abl, the electrophoretic force exerted on the mutants, and the wild-type protein should be the same. So the changes in the S1/S2 transition kinetics induced by the mutations cannot be attributed to the electrophoretic effect. If so, these mutated residues must be directly involved in forming the Abl/nanopore interactions, which could explain the weakened interactions in the Y320G and T319/Y320G and enhanced interactions in G321V mutants. Because removing hydrophobic Y320 impaired the interaction and introducing valine to G321 promoted the interaction, the result suggested that the hinge contributed to the Abl/nanopore interface via hydrophobic interactions. This would contradict several facts of the Abl and ClyA structure:

1. The hinge residues are located at a highly negative patch of Abl surface, as shown in the Figure below.
2. The G321V residue is located in a groove formed between the N and C lobes, far from the surface. The increased hydrophobicity of this position cannot contribute to the surface-surface interactions between Abl and ClyA.
3. The ClyA lumen is highly charged.

Thus, our interpretation of S1/S2 transition as the lobe open-close motion, the kinetic of which is associated with the flexibility of the hinge, is more reasonable and in line with previous works.

Here we also want to share a piece of unpublished data about Src kinase to further demonstrate our point. Both Src kinase and Abl belong to the tyrosine kinase family, and the two enzymes share 49% of sequence identity. The figure below shows that Src also exhibited the S1/S2 transition with S1 containing multiple sub-states. The similar current state patterns between the two functionally related proteins strongly suggest that the current states observed here were associated with their intrinsic motions. The result thus argues against the S1/S2 states originate from kinase moving around nanopore as the two kinases differ in 50% of the amino acid composition and have different surface charge distributions. It would be implausible that the two proteins with different surface properties still interact with ClyA nanopore in a similar manner. Detailed characterization of Src and comparison of

4. The authors should add longer traces (e.g. 20 sec) for the data showed in Figure 2a-f.

Reply: We have added several longer traces as shown in **Supplementary Fig. 8**.

5. Figure 3f. How can the trapping time be shorter than the dwell time of S2?

Reply: When calculating the dwell time of Abl proteins, all the trapping events were collected, and the distribution of the dwell times were fitted with a single exponential function to derive the average dwell time. Interestingly, we noticed that the trapping events mostly started with S1 signal and ended with S2 signal indicating that the Abl protein entered the nanopore in S1 state and escaped from ClyA while in S2 state (The reason for such phenomenon is to be investigated). Because of that, we believe the duration times of the initial S1 and the last S2 event of a trace were truncated. Many short Abl-trapped events contained only one S1-S2 transition cycle. Although they were included in the calculation of the average trapping time, these S1 and S2 events were excluded from the calculation of τ_{S1} and τ_{S2} . For calculating τ_{S1} and τ_{S2} , we only used the S1 and S2 events in the middle of long trapping traces that contained multiple turnovers of S1 and S2 states with the first S1 and the last S2 events discarded (See **Supplementary Figure 5**). Because of that, the τ_{S2} is longer than the average trapping time of Abl as the τ_{S2} was derived from those trapping events with long dwell times.

6. Page 4. Please indicate how I_{res} was calculated (e.g by indicating the open pore current and blocked pore current).

Reply: We added a new figure as **Supplementary Fig. 1** and updated the text in **page 4** as follows “Residual current ($I_{res}\%$) was calculated from blocked pore current (I_B) and open pore current (I_O) with $I_{res}\% = 100 * I_B / I_O$.”

7. Page 5. Figure 2d. Is S1d an independent state or simply S1b but shifted? Can this be proven?

Reply: If S1d is a shifted S1b (same nucleotide binding mode), we would expect the same relative occupancy of S1b and S1d state for ATP and ATPyS bound states at the saturated ATP/ATPyS concentration. However, there was an obvious difference in the S1b to S1d ratio among the two nucleotide bonding states (**Fig. 2d-e. Supplementary Table 4**). Therefore, we speculated there were two slightly different nucleotide binding modes with ATP slightly preferring the S1d state while ATPyS preferring the S1a. So far, we do not have any good experiment to prove this explanation.

8. Is the ternary complex sampled using ATP or gamma-ATP?

Reply: It is gamma-ATP. We've updated the text in **page 6** to clarify it as: “The ternary N_{4pos} Abl:ATP:Abltide complex was mimicked by an N_{4pos} Abl:ATPyS:Abltide state.”

9. The authors indicate a K_D . However, it is not really clear what is the meaning of the K_D . Is there any dependence on the applied potential (can be tested) or confinement inside the nanopore (should be briefly discussed)?

Reply: We've updated the text in **page 8** to clarify the definition of K_D as follows: “We also determined the apparent K_D of imatinib for binding to Abl, defined as the imatinib concentration at which half of the Abl population were loaded with the inhibitor.”

Earlier studies have shown that the K_D values derived from nanopore measurement could be comparable to bulk assays, or lower or higher than the reported values from bulk assays (<https://doi.org/10.1021/acsnano.9b09434>; <https://doi.org/10.1021/acsnano.9b07385>;

<https://doi.org/10.1021/jacs.5b01520>) Previous studies showed that voltages can affect the K_D of charged ligands to the analyte proteins, but not affect the K_D of neutral ligands to proteins. (<https://doi.org/10.1021/acsnano.9b09434>) We included this information in the revised manuscript as such (**page 12**) “. Note, previous ClyA nanopore tweezer studies have shown that the kinetics of ligand binding could be influenced by the applied potential and the confinement of nanopore imposed on trapped enzymes. It is well known that an electric field can stimulate protein motions. Given more than 25% of residues of the Abl kinase domain are charged, we anticipated that the electric field would have an impact on the structural dynamics of Abl within ClyA nanopore. Further ClyA nanopore tweezer analysis, in combine with molecule simulations will be carried out to explore how ligands, inhibitor molecules and drug-resistant mutations re-shape the conformational landscapes of Abl kinase, and to what extent the applied potential modulates the kinetics of these functional motions in the future.”

10. There is no evidence that the recognition is either conformational selection or induced fit. The authors should either elaborate on their interpretation or remove this sentence.

Reply: The definition of the conformational selection is that conformational changes occur at enzymes at the unbound state and ligand binding selectively stabilizes the pre-existing conformational states. Our data clearly showed that apo Abl visited S1a, b and c states that can be selectively stabilized by ATP, Abtide and ATP+Abtide binding. We think these data strongly support the conformational selection mechanism.

We proposed the induced fit mechanism based on the observation of S1d, a new state induced by ATP and ATP γ S. However, due to lack of solid evidence that S1d was a new conformational state, we decide to withdraw this statement and modify the text (**page 7**) as

“Also, the observation that apo Abl kinase can sample different S1a, S1b and S1c states which were selectively stabilized by different ligands, suggested a conformational selection mechanism for ligand recognition by Abl.”

11. Page 9-10. How were the ΔG calculated?

Reply: We've added the information in Method section in **page 14** as follows:

“The free energy difference between two states was calculated with $\Delta G = -RT \ln K_{eq}$. For example, $\Delta G_{S1, S2} = -RT \ln K_{eq} = -RT \ln (T_{S1} / T_{S2})$.”

12. Page 10. The authors correlate S1 and S2 with the altered dynamics of the mutated proteins. However, the authors argued earlier that the dynamics important for functions (NMR study) are those involving the S1a-c current levels rather S1 and S2. This is making the argument confusing. It appears more logic to compare the S1a-c current levels between mutants and related them to the NMR data (also mutants were tested there).

Reply: We did not argue that only S1a-c states are functional related. Instead, both global motions (lobe open and close) and local dynamics (re-arrangement of loops) are important for functions. For example, in an important work by Masterson et al, the lobe motions of protein kinase A were reported to dominate the rate-limiting step of kinase catalysis (<https://doi.org/10.1038/nchembio.452>). Thus, in our manuscript, we stated (**page 7**) “Conformational switch between lobe-open and -closed states have been observed in many protein kinases and shown to play an essential role in catalysis and kinase activation.” NMR study didn't report the lobe motions for Abl, which might be due to that the timescale of lobe motions were beyond the NMR method detection limit or the kinase was fixed at the lobe-closed state at the low 10 °C. Our nanopore measurements were able to observe both a slow S1/S2 motion and fast S1a/c fast transitions. As we stated in the previous Q&R session (Question 2), because the energy landscape of the conformational states and effect of mutants could be drastically

different at two different temperatures, the population distributions and kinetics of S1a-c in the nanopore measurements may not be quantitatively comparable with the NMR conformers.

Reviewer 2:

This is an interesting work that uses tweezers to look at the conformational ensemble of Abl kinase at the single-molecule level. Although by itself this approach cannot provide structural information, combined with the recent atomic insight into the various conformational states of Abl by NMR, it can be very informative. The findings are of interest, especially considering how challenging going it is to extract information on a protein that is rapidly interconverting between conformational states. If this approach can be further developed and refined it could be a very useful complement to NMR.

In the experiment with imatinib, the authors should bring to the attention of the reader that the drug can also bind allosterically to Abl (PMID 34774565). This is not an issue for the current work since the authors work at low concentration, but it is something that the authors need to discuss.

Reply: We thank the reviewer for the positive recommendation. We've added the allosteric binding information in **page 7** to notify readers as follows:

“Imatinib was shown to bind to ATP pocket by crystal structures (PDB: 2HYY, 1IEP, 3K5V) with a long residence time ($1/k_{\text{off}}$) of ~23 minutes, making it a useful tool to lock Abl kinase in a specific conformational state. Of note imatinib was recently shown to bind with Abl at the myristoyl pocket ($K_D = \sim 10 \mu\text{M}$) at high imatinib concentration with a much shorter residence time of ~10 ms.”

Reviewer 3:

“Mapping the conformational energy landscape of Abl kinase using ClyA nanopore tweezer” by Fanjun Li et al. described an approach using ClyA nanopore to investigate the conformational dynamics of Abelson (Abl) kinase. Two major conformational states of Abl (S1 and S2) and three sub-states of S1 have been characterized (S1a, S1b, and S1c). Moreover, the authors investigated the interaction of ligands, substrates with N4posAbl (Abl with a positively charged peptide tail at its N-terminus), suggesting that sub-states of S1 reflect the ligand/substrate binding conformations while S2 state is non-ligand conformation. Several mutants have been designed to check the S1/S2 transition kinetics and their interaction with substrates. Overall, this is a very interesting manuscript reporting an important development in the nanopore field. The results obtained in this study have a potential impact on applying nanopores to study the conformational dynamics of proteins. However, there are some questions that need to be addressed before publication.

Reply: We thank the reviewer for the positive recommendation.

Major:

1. In the introduction part (Page 2), the authors stated “...but it remains challenging to obtain state transition kinetics owing to the large system size and long timescales associated with kinase conformational dynamics.” Could the authors explain why the large system size and long timescales conformational dynamics make it challenging to obtain state transition kinetics?

Reply: Traditional MD simulations is limited to timesteps of about 2 fs due to high-frequency resonance frequencies. Conformational changes in protein kinases occur on the microsecond to millisecond range (<http://dx.doi.org/10.1016/j.tibs.2015.09.002>), which is 9-12 orders of magnitude greater than the timesteps possible with traditional MD. Further, each step requires a costly force calculation. As such, simulating medium-size proteins on a large, distributed cluster often requires months of computer time to simulate milliseconds of dynamics, while simulating a large protein (e.g.,

the β -2 adrenergic receptor) on biologically relevant time scales (milliseconds through hours) using a standard desktop computer would take years. (<https://doi.org/10.1021/ct400331r>)

NMR study of DFG-flipping in Abl kinase suggested a timescale of millisecond. (DOI: [10.1126/science.abc2754](https://doi.org/10.1126/science.abc2754)) A recent computational study used Markov state models constructed from a total of 800 μ s of aggregate molecular dynamics data to explore the free energy landscape of *apo* Abl. (<https://doi.org/10.1021/acs.jctc.9b01158>) As the authors themselves pointed out that the timescale of this simulation didn't cover the conformational changes of Abl kinase, thus the possible conformations or the transition kinetics may not be accurately determined.

We updated the text (**page 2**) as “but it remains challenging to characterize the large conformation changes occurring over long timescales (μ s ~ ms) owing to the large protein size and the small calculation timesteps (~2 fs).”

Could the authors comment/introduce the currently available methods/studies that investigated the conformational dynamics of Abelson kinase?

Reply: Computational simulations and experimental methods including MS and NMR have been used to investigate the conformational dynamics of Abl kinase. We've updated the text in **page 2**.

2. The authors stated “...(ClyA) nanopore variants to reveal their ligand bound and catalytic states. Notably, four different ionic states of dihydrofolate reductase (DHFR) were detected³⁴ and three anomeric maltose-bound states of maltose binding protein (MBP) were resolved by ClyA nanopores...” Could the authors comment on what's the difference or highlight between their manuscript and the papers that have been cited here since these studies already have proved that ClyA nanopores are able to detect different ligand bindings, catalytic states and three anomeric binding states of target proteins?

Reply: The structural dynamics of MBP and DHFR were intensively studied via crystallography and NMR. Previous ClyA nanopore studies used these two proteins as model systems to develop and establish the utility of the nanopore tweezer method for monitoring the structural dynamics. Due to the high flexibility, the detailed conformational dynamics of Abl kinase has not been well characterized. Therefore, we applied the nanopore tweezers technique to study this highly medically relevant protein. We hope our work, complementary to the exiting structural studies, would provide new information to the kinase field, particularly the state transition kinetics which are challenging to be extracted by other methods.

3. Page 3, Fig. 1 caption “...The current traces were collected at -80 mV in 150 mM NaCl, 100 mM Tris-HCl, pH 7.5...” Could the author explain why -80 mV has been chosen for collecting the data? I would assume the application of voltage may have some effect on the trapping time of protein inside the nanopore and therefore possibly affect the kinetics, especially the authors introduced a positively charged tag in the protein here. Could the authors show the relationship between trapping time and different voltages of Abl, AblC4pos and N4posAbl, for instance, -60, -70, -80, -90, -100mV, respectively.

Reply: Thank you for the suggestion. We performed the voltage dependent trapping of N₄posAbl and found that -80mV enabled the longest trapping time for accurately calculation of the transition time of S1 and S2. New **Supplementary Figure 3** and **Supplementary Table 2** are now included to show this result.

4. Page 4, "...two Abl constructs with a positively charged peptide tail at its N-terminus, termed N4posAbl, or to its C-terminus, named AblC4pos. Both AblC4pos and N4posAb; showed longer trapping of 1.52 ± 0.13 s (N=3) and 20.72 ± 2.83 s (N=3), respectively..." Could the authors show what's the sequence of positively charged peptide tail?

Reply: We've updated the sequence information in **page 4** as follows:
"peptide sequence for N4pos tag: KRKKS GG; C4pos tag: GGSKRK"

Although the authors described "...plasmid pET_His10 TEV" in the method part, it's not clear;

Reply: We've updated the plasmid information in **Materials and Methods (page 12)** as

"pET_His10 TEV_Abl1 kinase domain (residue 229-512)"

the authors stated the data from 3 different pores but didn't mention how many events have been collected from each pore to get statistic results.

Reply: We've added the event numbers.

It's also interesting that AblC4pos and N4posAbl show a very different trapping, N4posAb is almost 200 times longer than AblC4pos, could the authors comment about the possible reasons of such big difference?

Reply: We are also puzzled by this observation as Abl_{C4pos} and N_{4pos}Abl share the same net charge. One possibility is that the C4pos tag changed the dipole of kinase. The Abl_{C4pos} and N_{4pos}Abl structures were modeled with Chimera using the x-ray crystallographic structure 2HYY from protein data bank. The dipole moments of those structures (Abl, Abl_{C4pos} and N_{4pos}Abl) were predicted by an online server: <https://dipole.proteopedia.org/>, as shown below:

The change of dipole moment in Abl_{C4pos} may result in an orientation opposite to N_{4pos}Abl inside of ClyA. Consequently, the interaction interface formed between Abl_{C4pos} and ClyA is likely to be different from that between N_{4pos}Abl and ClyA. This could explain the drastically different dwell time between the two Abl kinase. We are collaborating with a computational biophysical group to set up MD simulations for the ClyA/Abl system to gain more insight into this phenomenon. In the revised manuscript (**page 4**), we added the following text to provide some explanations for the observation:

“We surmised that the two kinase proteins may enter ClyA in different orientations with $N_{4\text{pos}}\text{Abl}$ binding to the ClyA lumen stronger than $\text{Abl}_{\text{C4pos}}$.”

And why AblC4pos trapping time is not long enough to study the conformational energy landscape of Abl.

Reply: Due the short dwell time, the $\text{Abl}_{\text{C4pos}}$ traces showed only one or less than one cycle of S1 and S2 events. We have observed that the trapping traces always started with S1 signal and exit with S2 signal. Under this condition, the duration of S1 and S2 events were truncated due the entering or escaping of the $\text{Abl}_{\text{C4pos}}$ protein. Therefore, we cannot accurately determine the S1/S2 transition kinetics for this construct.

5. Page 4, the authors stated “...In current recording experiments, $N_{4\text{pos}}\text{Abl}$ induced similar current signals as Abl (Fig. 1c-e, Supplementary Fig. 3).” However, why there is no data for Abl in Supplementary Table 2? According to Fig. 1c, Abl did show S1 and S2 states, please add it.

Reply: It is true that Abl showed S1 and S2 states. However, “in calculating τ_{S1} and τ_{S2} , the S1 or S2 states at the beginning and those at the end of a trapping event were discarded as their true dwell times were interrupted by Abl entering and exiting the ClyA-AS nanopore (see **Supplementary Fig. 5**.)” (the text can be found in our Method section, **page 14**). The trapping time of untagged Abl is so short that not many complete S1 and S2 events were observed, therefore we were not able to determine the S1/S2 transition kinetics with this construct.

6. The Michaelis-Menten kinetics analysis of AblC4pos is missing, please add it in Supplementary Table 1 and Supplementary Figure 1.

Reply: We’ve added the Michaelis-Menten kinetics analysis of $\text{Abl}_{\text{C4pos}}$ to **Supplementary Table 1** and made a new **Supplementary Figure 2**.

7. Page 6, “...mutations were introduced into the hinge region (Phe317-Leu323) of Abl11 to generate two mutants ($N_{4\text{pos}}\text{Y320G}$ and $N_{4\text{pos}}\text{T319G Y320G}$) with increased flexibility, and one mutant ($N_{4\text{pos}}\text{G321V}$) with enhanced rigidity”. Could the authors refer to some references or use some structure data to demonstrate that mutated proteins here ($N_{4\text{pos}}\text{Y320G}$ and $N_{4\text{pos}}\text{T319G Y320G}$) do have increased flexibility while $N_{4\text{pos}}\text{G321V}$ shows an enhanced rigidity. Ref.11 didn’t study these mutants.

Reply: We didn’t find specific studies on the hinge flexibility of Abl kinase. But there are two related studies on the extracellular signal-regulated kinase 2 (ERK2), in which increasing hinge mobility of ERK2 by introducing Gly mutations at the hinge promoted domain motion within the kinase core (<https://doi.org/10.1073/pnas.1318899111>; doi: [10.1016/j.jmb.2014.02.011](https://doi.org/10.1016/j.jmb.2014.02.011)). We deleted the Ref.11 to avoid the confusion and cited the two references on ERK2. The text (**page 7**) is revised as follows: “Studies on the extracellular signal-regulated kinase 2 (ERK2) suggested that the activation of ERK2 was promoted by substitution of the hinge domain residues with Gly to increase the flexibility of the hinge region that controls the N- and C-lobe domain movement.^{47, 48} To probe if S1/S2 states transition is related to the domain movement, we also introduced Gly into the hinge region (Phe³¹⁷-Leu³²³) of Abl to generate two mutants ($N_{4\text{pos}}\text{Y320G}$ and $N_{4\text{pos}}\text{T319G Y320G}$) with increased flexibility, as well as one mutant ($N_{4\text{pos}}\text{G321V}$) that intended to enhance rigidity (**Fig. 3a**).”

8. If I'm not mistaken, the trace shown in Fig 2a, Fig. 4b and Fig. 5a are exactly the same. It will be just nice to see another trace of N4posAbl trapped within ClyA.

Reply: We have replaced **Fig. 4b** and **Fig. 5a** with different traces of N4posAbl.

9. Page 10, "...while still maintain a high temporal resolution at 100 μ s", could the authors explain why the resolution is 100 μ s according to their description in method (Page 13) "...at a sampling rate of 50 kHz after processing with a 4-pole lowpass Bessel filter at 2 kHz".

Reply: We meant that the temporal resolution of 100 μ s is technically achievable by the instrument achieve although we didn't perform the experiments at 10 kHz. To avoid the confusion, we rephrase the statement (**page 12**) as "Here we have demonstrated nanopore tweezers that can measure structural dynamics at a time scale (ms ~ 10s). Note recordings at a broader time scale (100 μ s ~ hr) while still maintain a high temporal resolution at 100 μ s are easily achievable with commercial electrophysiological instruments."

10. Page 13, data analysis, how did the authors process the data? Clampfit or customized script? How did the author define peak S1a, b, c and d? For instance, the histogram shown in Fig .1c is actually in between S1b and S1c. Fig. 1d-e, how the authors assign the overlap part between S1d and S1b?

Reply: Clampfit was used for data analysis. We included this information in the Method (**page 14**). We used the peaks in the current histograms to distinguish the states (S1a-d). S1d was defined independent from S1a or S1b because all three peaks distinctly exist in the ATP/ATP γ S saturating condition, while there is no overlapping S1d peak in the apo state (**Fig. 2g**). In the saturating condition for Abltide, there is only one dominant peak which falls within the Ires% range of S1b^{apo} and although it is slightly shifted away from S1b^{apo}, it is considered S1b state. Besides, the Abltide binding state of N4posG321V resembled S1b state (**Fig. 5d**).

Minor:

1. Page 2, "...to characterize the dynamics A-loop", maybe "to characterize the dynamics of Aloop" is better?

Reply: We really appreciate the reviewer for reading our manuscript so carefully! The mistake was corrected.

2. Page 4, "...Both AblC4pos and N4posAb; showed longer..." contains an apparent typo

Reply: The text has been updated in **page 4** as "Both Abl_{C4pos} and N4posAbl showed..."

3. Page 13, "State population (P) was calculated either by state dwell times, for example, $PS1 = \tau S1 / (\tau S1 + \tau S2)$. Relative state occupancy in the presence of different ligands was calculated by peak area in histograms..." contains an uncompleted sentence and a typo.

Reply: The text has been updated in **page 14** as "State population (P) was calculated by state dwell times, for example, $P_{S1} = \tau_{S1} / (\tau_{S1} + \tau_{S2})$. Relative state occupancy in the presence of different ligands was calculated by peak area in histograms."

REVIEWERS' COMMENTS

Reviewer #1 (Remarks to the Author):

The authors have greatly improved the manuscript. It is a pity, they could not perform an voltage dependency of the current blockades. This would have allowed understanding the energy profile of the protein inside the nanopore and show a more convincing correlation between the ionic signal and the intrinsic dynamics of the protein. Nevertheless, the authors have argued rather convincingly with other experimental evidences that the current blockades are indeed reflecting the intrinsic dynamics of the protein inside the nanopore.

The authors have also improved the flow of the manuscript. They comparison between the NMR data and the current recordings it is now at the end of the manuscript rather than at the beginning. This is wise, as it seems that the correlation between NMR and nanopore experiments is not straight forward. As it stands the flow of the manuscript is more logical and the reader is not confused by the meaning of the NMR experiments.

A small comment, On page 4, the authors should define the I_{res} in relation to the open pore and blocked pore currents.

Reviewer #3 (Remarks to the Author):

The authors have addressed all points raised in the previous round of review.

I just realized I did a mistake in my previous review about the last Major comment. I'm really sorry for it. In fact, I mean Fig. 2 rather than Fig.1. Therefore, the comment should be "...How did the author define peak S1a, b, c, and d? For instance, the histogram shown in Fig .2c is actually in between S1b and S1c. Fig. 2d-e, how do the authors assign the overlap part between S1d and S1b?" Could the authors describe these details? If the authors use I_{res} to define them, could the authors show in the method or main text from which rang is assigned for S1a, S1b, S1c, and S1d, respectively? Also, could the authors describe how they get the "Trapping time" value, is it from a fit of dwell time histogram, or is it averaged dwell time?

Reply to Reviewers

We thank the reviewers for their positive recommendation of our manuscript. In the revised manuscript, we have addressed the issues raised by the reviewers, with major changes highlighted in yellow.

REVIEWERS' COMMENTS

Reviewer #1 (Remarks to the Author):

The authors have greatly improved the manuscript. It is a pity, they could not perform an voltage dependency of the current blockades. This would have allowed understanding the energy profile of the protein inside the nanopore and show a more convincing correlation between the ionic signal and the intrinsic dynamics of the protein. Nevertheless, the authors have argued rather convincingly with other experimental evidences that the current blockades are indeed reflecting the intrinsic dynamics of the protein inside the nanopore.

The authors have also improved the flow of the manuscript. They comparison between the NMR data and the current recordings it is now at the end of the manuscript rather than at the beginning. This is wise, as it seems that the correlation between NMR and nanopore experiments is not straight forward. As it stands the flow of the manuscript is more logical and the reader is not confused by the meaning of the NMR experiments.

Reply: We thank the reviewer for the positive recommendation.

A small comment, On page 4, the authors should define the I_{res} in relation to the open pore and blocked pore currents.

Reply: We've updated the text (page 3, highlighted in yellow) as:

"Residual current (I_{res}) was calculated from blocked pore current (I_B) and open pore current (I_O) with $I_{res}\% = 100\% \times I_B/I_O$."

Reviewer #3 (Remarks to the Author):

The authors have addressed all points raised in the previous round of review.

I just realized I did a mistake in my previous review about the last Major comment. I'm really sorry for it. In fact, I mean Fig. 2 rather than Fig.1. Therefore, the comment should be "...How did the author define peak S1a, b, c, and d? For instance, the histogram shown in Fig .2c is actually in between S1b and S1c. Fig. 2d-e, how do the authors assign the overlap part between S1d and S1b?" Could the authors describe these details? If the authors use $I_{res}\%$ to define them, could the authors show in the method or main text from which rang is assigned for S1a, S1b, S1c, and S1d, respectively? Also, could the authors describe how they get the "Trapping time" value, is it from a fit of dwell time histogram, or is it averaged dwell time?

Reply: We used the current range of peaks in the histogram to define the S1 substates. Specifically, the $I_{res}\%$ range of N4posAbl S1 substates under different conditions at -80 mV are

defined as S1a (49 ~ 64%), S1b (28 ~ 49%), S1c (15 ~ 28%), S1d (40 ~ 56%). At apo state, the S1a, S1b, and S1c are clearly distinguishable by Ires%. However, in the presence of ATP or ATPyS, a new state S1d could be visually recognized and distinguished from S1a and S1b by its characteristic noise pattern. However, the current range of the new state S1d overlapped largely with S1a and S1b making it difficult to use Clampfit to precisely determine the dwell time of each individual state by Ires%. While in this manuscript, we have not discussed the kinetics of the ligand binding in detail, we are working on developing our own program that aims to define the binding events by considering multiple parameters, including current level, the rmsd etc. We hope sometime soon we will be able to report the findings about the ligand interaction once our program can accomplish the task.

We updated the text in Methods (page 10, highlighted in yellow) as:

“The current range of peaks in the histogram was used to define the S1 substates. Specifically, the Ires% range of N4posAbl S1 substates under different conditions at -80 mV are defined as S1a (49 ~ 64%), S1b (28 ~ 49%), S1c (15 ~ 28%), S1d (40 ~ 56%). At apo state, the S1a, S1b, and S1c are clearly distinguishable by Ires% while the nucleotide binding generated a new S1d state whose current distributions largely overlapped with S1a and S1b.”

S1d was defined independent from S1a or S1b because all three peaks distinctly exist in the ATP/ATPyS saturating condition, while there is no overlapping S1d peak in the apo state (**Fig. 2g**). In the saturating condition for Abltide, there is only one dominant peak which falls within the Ires% range of S1b^{apo} and although it is slightly shifted away from S1b^{apo}, it is considered S1b state. Besides, the Abltide binding state of N4posG321V resembled S1b state (**Fig. 5d**).

The reported trapping time is an averaged value of dwell times derived from single exponential fitting of the event duration histograms of at least three independent pore measurements. We updated the text (page 10, highlighted in yellow) as:

“Dwell times of trapping events and current states were determined by using single-channel search in Clampfit 11.1. All dwell times from one nanopore experiment were binned and fitted to a single-exponential function to derive the τ . The average dwell times and the standard deviation were determined from at least three independent nanopore measurements.”